# Unsupervised-to-Online Reinforcement Learning

## Abstract

Offline-to-online reinforcement learning (RL), a framework that trains a policy with offline RL and then further fine-tunes it with online RL, has been considered a promising recipe for data-driven decision-making. While sensible, this framework has drawbacks: it requires domain-specific offline RL pre-training for each task, and is often brittle in practice. In this work, we propose **unsupervised-to-online RL** (**U2O RL**), which replaces domain-specific *supervised* offline RL with *unsupervised* offline RL, as a potentially better alternative to offline-to-online RL. U2O RL not only enables reusing a single pre-trained model for multiple downstream tasks, but also learns better representations, which often result in *even better* performance and stability than *supervised* offline-to-online RL. To instantiate U2O RL in practice, we propose a general recipe for U2O RL to bridge task-agnostic unsupervised offline skill-based policy pre-training and supervised online fine-tuning. Throughout our experiments in eleven state-based and pixel-based environments, we empirically demonstrate that U2O RL often achieves strong performance that matches or even outperforms previous offline-to-online RL approaches when the dataset consists of diverse trajectories, while being able to reuse a single pre-trained model for a number of different downstream tasks.

## 1 Introduction

Across natural language processing (NLP), computer vision (CV), and speech processing, ubiquitous in the recent successes of machine learning is the idea of adapting an expressive model pre-trained on large-scale data to domain-specific tasks via fine-tuning. In the domain of reinforcement learning (RL), offline-to-online RL has been considered an example of such a recipe for leveraging offline data for efficient online fine-tuning. Offline-to-online RL first trains a task-specific policy on a previously collected dataset with offline RL, and then continues training the policy with additional environment interactions to further improve performance.

But, is offline-to-online RL really the most effective way to leverage offline data for online RL? Offline-to-online RL indeed has several limitations. First, it pre-trains a policy with a *domain-specific task reward*, which precludes sharing a single pre-trained model for multiple downstream tasks. This is in contrast to predominant pre-training recipes in large language models or visual representation learning, where they pre-train large models with self-supervised or *unsupervised* objectives to learn useful representations, which can facilitate learning a wide array of downstream tasks. Second, naïve offline-to-online RL is often brittle in practice (Lee et al., 2022; Nakamoto et al., 2023). This is mainly because pre-trained offline RL agents suffer the distributional shift between the offline and online interaction data (Lee et al., 2022; Nakamoto et al., 2023) or experience feature collapse (Section 5), which necessitates specialized, potentially complicated techniques.

In this work, our central hypothesis is that *unsupervised pre-training of diverse policies* from offline data can serve as an effective data-driven recipe for *online* RL, and can be more effective than even domain-specific ("supervised") offline pre-training. We call this recipe **unsupervised-to-online RL** (**U2O RL**). U2O RL has two appealing properties. First, unlike offline-to-online RL, a single pre-trained model can be fine-tuned for different downstream tasks. Since offline unsupervised RL does not require task information, we can pre-train diverse policies on unlabeled data before knowing downstream tasks. Second, by pre-training multi-task policies with diverse intrinsic rewards, the agent extracts rich representations from data, which often helps achieve *even better* final performance

**Offline-to-Online RL**

(Supervised) Offline RL Pre-training

Online RL Fine-tuning

$(s, a, r, s')$

$\pi(a \mid s)$

$(s, a, r, s')$

$\pi(a \mid s)$

**Unsupervised-to-Online RL**

Unsupervised Offline RL Pre-training

Bridging

Online RL Fine-tuning

$(s, a, s', r^{\text{int}}, z)$

$z \in \mathcal{Z}$

$\pi(a \mid s, z)$

$(s, a, r, s')$

$\pi(a \mid s, z^*)$

$(s, a, r, s')$

$\pi(a \mid s, z^*)$

Figure 1: **Illustration of U2O RL.** In this work, we propose to replace supervised offline RL with *unsupervised offline RL* in the offline-to-online RL framework. We call this scheme **unsupervised-to-online RL** (**U2O RL**). U2O RL consists of three stages: (1) unsupervised offline RL pre-training, (2) bridging, and (3) online RL fine-tuning. In unsupervised offline RL pre-training, we train a multi-task skill policy $\pi_\theta(a \mid s, z)$ instead of a single-task policy $\pi_\theta(a \mid s)$. Then, we convert the multi-task policy into a task-specific policy in the bridging phase. Finally, we fine-tune the skill policy with online environment interactions.

and stability than *supervised* offline-to-online RL. This resembles how general-purpose unsupervised pre-training in other domains, such as with LLMs or self-supervised representations (Brown et al., 2020; Devlin et al., 2019; Radford et al., 2019; He et al., 2021; 2020; Hénaff et al., 2020), improves over the performance of domain-specific specialist pre-training.

U2O RL consists of three stages: unsupervised offline policy pre-training, bridging, and online fine-tuning (Figure 1). In the first unsupervised offline pre-training phase, we employ a skill-based offline unsupervised RL or offline goal-conditioned RL method, which trains diverse behaviors (or *skills*) with intrinsic rewards and provides an efficient mechanism to identify the best skill for a given task reward. In the subsequent bridging and fine-tuning phases, we adapt the best skill among the learned policies to the given downstream task reward with online RL. Here, to prevent a potential mismatch between the intrinsic and task rewards, we propose a simple yet effective reward scale matching technique that bridges the gap between the two training schemes and thus improves performance and stability.

Our main contributions in this work are twofold. First, to the best of our knowledge, this is the first work that makes the (potentially surprising) observation that it is often better to replace supervised offline RL with unsupervised offline RL in the offline-to-online RL setting. We also identify the reason behind this phenomenon: this is mainly because offline unsupervised pre-training learns better *representations* than task-specific supervised offline RL. Second, we propose a general recipe to bridge skill-based unsupervised offline RL pre-training and online RL fine-tuning. Through our experiments on eleven state-based and pixel-based environments, we demonstrate that U2O RL often outperforms standard offline-to-online RL both in terms of sample efficiency and final performance, while being able to reuse a single pre-trained model for multiple downstream tasks.

## 2 RELATED WORK

**Online RL from prior data**. Prior works have proposed several ways to leverage a previously collected offline dataset to accelerate online RL training. They can be categorized into two main groups: offline-to-online RL and off-policy online RL with offline data. Offline-to-online RL first pre-trains a policy and a value function with offline RL (Lange et al., 2012; Levine et al., 2020; Fujimoto & Gu, 2021; Fujimoto et al., 2019; Kumar et al., 2019; Tarasov et al., 2023a; Wu et al., 2019a; Kostrikov et al., 2021; Kumar et al., 2020; Hansen-Estruch et al., 2023; Kostrikov et al., 2022; Nair et al., 2020; Peng et al., 2019; Wang et al., 2020), and then continues to fine-tune them with additional online interactions (Lee et al., 2022; Nair et al., 2020; Nakamoto et al., 2023; Yu & Zhang, 2023; Lei et al., 2023; Zheng et al., 2022; Mark et al., 2022; Zhao et al., 2023). Since naïve offline-to-online RL is often unstable in practice due to the distributional shift between the dataset and

online interactions, prior works have proposed several techniques, such as balanced sampling (Lee et al., 2022), actor-critic alignment (Yu & Zhang, 2023), adaptive conservatism (Wang et al., 2023a), and return lower-bounding (Nakamoto et al., 2023). In this work, unlike offline-to-online RL, which trains a policy with the target task reward, we offline pre-train a multi-task policy with unsupervised (intrinsic) reward functions. This makes our single pre-trained policy reusable for any downstream task and learn richer representations. The other line of research, off-policy online RL, trains an online RL agent from scratch on top of a replay buffer filled with offline data, without any pre-training (Ball et al., 2023; Li et al., 2023; Luo et al., 2024; Song et al., 2023). While this simple approach often leads to improved stability and performance (Ball et al., 2023), it does not leverage the benefits of pre-training; in contrast, we do leverage pre-training by learning useful features via offline unsupervised RL, which we show leads to better fine-tuning performance in our experiments.

**Unsupervised RL.** The goal of unsupervised RL is to leverage unsupervised pre-training to facilitate downstream reinforcement learning. Prior works have mainly focused on unsupervised representation learning and unsupervised behavior learning. Unsupervised representation learning methods (Sermanet et al., 2018; Shah & Kumar, 2021; Parisi et al., 2022; Xiao et al., 2022; Nair et al., 2022; Ma et al., 2023b;a; Ghosh et al., 2023; Seo et al., 2022b;a; 2023) aim to extract useful (visual) representations from data. These representations are then fed into the policy to accelerate task learning. In this work, we focus on unsupervised behavior learning, which aims to pre-train policies that can be directly adapted to downstream tasks. Among unsupervised behavior learning methods, online unsupervised RL pre-trains useful policies by either maximizing state coverage (Pathak et al., 2017; 2019; Mendonca et al., 2021; Liu & Abbeel, 2021) or learning distinct skills (Gregor et al., 2016; Eysenbach et al., 2019b; Sharma et al., 2020; Park et al., 2024d) via reward-free interactions with the environment. In this work, we consider *offline* unsupervised RL, which does not allow any environment interactions during the pre-training stage.

**Offline unsupervised RL.** Offline unsupervised RL methods focus on learning diverse policies (*i.e.*, skills) from the dataset, rather than exploration, as online interactions are not permitted in this problem setting. There exist three main approaches to offline unsupervised RL. Behavioral cloning-based approaches extract skills from an offline dataset by training a generative model (*e.g.*, variational autoencoders (Kingma & Welling, 2014), Transformers (Vaswani et al., 2017), etc.) (Ajay et al., 2021; Pertsch et al., 2021; Singh et al., 2021). Offline goal-conditioned RL methods learn diverse goal-reaching behaviors with goal-conditioned reward functions (Chebotar et al., 2021; Eysenbach et al., 2022; Ma et al., 2022; Park et al., 2024b; Wang et al., 2023b; Yang et al., 2023; Fang et al., 2022; 2023). Offline unsupervised skill learning approaches learn diverse skills based on intrinsically defined reward functions (Park et al., 2024c; Touati et al., 2022; Hu et al., 2023). Among these approaches, we use methods in the second and third categories (*i.e.*, goal- or skill-based unsupervised offline RL) as part of our method.

Our goal in this work is to study how unsupervised offline RL, as opposed to supervised task-specific offline RL, can be employed to facilitate online RL fine-tuning. While somewhat similar unsupervised pre-training schemes have been explored in prior works, they either consider hierarchical RL (or zero-shot RL) with frozen learned skills without fine-tuning (Ajay et al., 2021; Pertsch et al., 2021; Touati et al., 2022; Park et al., 2024c; Hu et al., 2023), assume online-only RL (Laskin et al., 2021), or are limited to the specific setting of goal-conditioned RL (Fang et al., 2022; 2023; Eysenbach et al., 2019a; Nasiriany et al., 2019). To the best of our knowledge, this is the first work that considers the *fine-tuning* of skill policies pre-trained with unsupervised offline RL *in the context of offline-to-online RL*. Through our experiments, we show that our fine-tuning framework leads to significantly better performance than previous approaches based on hierarchical RL, zero-shot RL, and standard offline-to-online RL.

## 3 PRELIMINARIES

We formulate a decision making problem as a Markov decision process (MDP) (Sutton & Barto, 2018), which is defined by a tuple of $(\mathcal{S}, \mathcal{A}, P, r, \rho, \gamma)$, where $\mathcal{S}$ is the state space, $\mathcal{A}$ is the action space, $P\colon \mathcal{S} \times \mathcal{A} \to \Delta(\mathcal{S})$ is the transition dynamics, $r\colon \mathcal{S} \times \mathcal{A} \times \mathcal{S} \to \mathbb{R}$ is the task reward function, $\rho \in \Delta(\mathcal{S})$ is the initial state distribution, and $\gamma \in (0, 1)$ is the discount factor. Our aim is to learn a policy $\pi\colon \mathcal{S} \to \Delta(\mathcal{A})$ that maximizes the expectation of cumulative task rewards, $\mathbb{E}_\pi[\sum_{t=0}^\infty \gamma^t r(s_t, a_t, s_{t+1})]$.

**Offline RL and implicit Q-learning (IQL).** The goal of offline RL is to learn a policy solely from an offline dataset $\mathcal{D}_{\texttt{off}}$, which consists of transition tuples $(s, a, s', r)$. One straightforward approach to offline RL is to simply employ an off-policy RL algorithm (*e.g.* TD3 (Fujimoto et al., 2018)). For instance, we can minimize the following temporal difference (TD) loss to learn an action-value function (Q-function) from data:

$$\mathcal{L}_{\texttt{TD}}(\phi) = \mathbb{E}_{(s,a,s',r)\sim\mathcal{D}_{\texttt{off}}} \left[ (r + \gamma \max_{a'} Q_{\bar{\phi}}(s', a') - Q_\phi(s, a))^2 \right], \tag{1}$$

where $Q_\phi$ denotes the parameterized action-value function, and $Q_{\bar{\phi}}$ represents the target action-value function (Mnih et al., 2013), whose parameter $\bar{\phi}$ is updated via Polyak averaging (Polyak & Juditsky, 1992) using $\phi$. We can then train a policy $\pi$ to maximize $\mathbb{E}_{a\sim\pi}[Q_\phi(s, a)]$.

While this simple off-policy TD learning can be enough when the dataset has sufficiently large state-action coverage, offline datasets in practice often have limited coverage, which makes the agent susceptible to value overestimation and exploitation, as the agent cannot get corrective feedback from the environment (Levine et al., 2020). To address this issue, Kostrikov et al. (2022) proposed implicit Q-learning (IQL), which fits an optimal action-value function without querying out-of-distribution actions: IQL replaces the $\arg\max$ operator, which potentially allows the agent to exploit Q-values from out-of-distribution actions, with an expectile loss that implicitly approximates the maximum value. Specifically, IQL minimizes the following losses:

$$\mathcal{L}_{\texttt{IQL}}^Q(\phi) = \mathbb{E}_{(s,a,s',r)\sim\mathcal{D}_{\texttt{off}}} \left[ (r + \gamma V_\psi(s') - Q_\phi(s, a))^2 \right], \tag{2}$$

$$\mathcal{L}_{\texttt{IQL}}^V(\psi) = \mathbb{E}_{(s,a)\sim\mathcal{D}_{\texttt{off}}} \left[ \ell_\tau^2(Q_{\bar{\phi}}(s, a) - V_\psi(s)) \right], \tag{3}$$

where $Q_\phi$ and $Q_{\bar{\phi}}$ respectively denote the action-value and target action-value functions, $V_\psi$ denotes the value function, $\ell_\tau^2(x) = |\tau - \mathbb{1}(x < 0)|x^2$ denotes the expectile loss (Newey & Powell, 1987) and $\tau$ denotes the expectile parameter. Intuitively, the asymmetric expectile loss in Equation 3 makes $V_\psi$ implicitly approximate $\max_a Q_{\bar{\phi}}(s, a)$ by penalizing positive errors more than negative errors.

**Hilbert foundation policy (HILP).** Our unsupervised-to-online recipe requires an offline unsupervised RL algorithm that trains a skill policy $\pi_\theta(a \mid s, z)$ from an unlabeled dataset, and we mainly use HILP (Park et al., 2024c) in our experiments. HILP consists of two phases. In the first phase, HILP trains a feature network $\xi \colon \mathcal{S} \to \mathcal{Z}$ that embeds temporal distances (*i.e.*, shortest path lengths) between states into the latent space by enforcing the following equality:

$$d^*(s, g) = \|\xi(s) - \xi(g)\|_2 \tag{4}$$

for all $s, g \in \mathcal{S}$, where $d^*(s, g)$ denotes the temporal distance (the minimum number of steps required to reach $g$ from $s$) between $s$ and $g$. In practice, given the equivalence between goal-conditioned values and temporal distances, $\xi$ can be trained with any offline goal-conditioned RL algorithm (Park et al., 2024b) (see Park et al. (2024c) for further details). After training $\xi$, HILP trains a skill policy $\pi_\theta(a \mid s, z)$ with the following intrinsic reward using an off-the-shelf offline RL algorithm (Kostrikov et al., 2022; Fujimoto et al., 2018):

$$r^{\texttt{int}}(s, a, s', z) = (\xi(s') - \xi(s))^\top z, \tag{5}$$

where $z$ is sampled from the unit ball, $\{ z \in \mathcal{Z} : \|z\| = 1 \}$. Intuitively, Equation 5 encourages the agent to learn behaviors that move in every possible latent direction, resulting in diverse state-spanning skills (Park et al., 2024c). Note that Equation 5 can be interpreted as the inner product between the task vector $z$ and the feature vector $f(s, a, s') := \xi(s') - \xi(s)$ in the successor feature framework (Dayan, 1993; Barreto et al., 2017).

## 4 UNSUPERVISED-TO-ONLINE RL (U2O RL)

Our main hypothesis in this work is that task-agnostic offline RL pre-training of *unsupervised* skills can be more effective than task-specific, supervised offline RL for online RL fine-tuning. We call this recipe **unsupervised-to-online RL** (**U2O RL**). In this section, we first describe the three stages of U2O RL (Figure 1): unsupervised offline policy pre-training (Section 4.1), bridging (Section 4.2), and online fine-tuning (Section 4.3). We then explain why *unsupervised*-to-online RL is potentially better than standard *supervised* offline-to-online RL (Section 4.4).

## 4.1 Unsupervised offline policy pre-training

In the first unsupervised offline policy pre-training phase (Figure 1 (bottom left)), we train diverse policies (or *skills*) with intrinsic rewards to extract a variety of useful behaviors as well as rich features from the offline dataset $\mathcal{D}_{\texttt{off}}$. In other words, instead of training a single-task policy $\pi_\theta(a \mid s)$ with task rewards $r(s, a, s')$ as in standard offline-to-online RL, we train a *multi-task* skill policy $\pi_\theta(a \mid s, z)$ with a family of unsupervised, *intrinsic* rewards $r^{\texttt{int}}(s, a, s', z)$, where $z$ is a skill latent vector sampled from a latent space $\mathcal{Z} = \mathbb{R}^d$. Even if $\mathcal{D}_{\texttt{off}}$ contains reward labels, we do not use any reward information in this phase.

Among existing unsupervised offline policy pre-training methods (Section 2), we opt to employ successor feature-based methods (Dayan, 1993; Barreto et al., 2017; Wu et al., 2019b; Touati et al., 2022; Park et al., 2024c) or offline goal-conditioned RL methods (Chebotar et al., 2021; Park et al., 2024b) for our unsupervised pre-training, since they provide a convenient mechanism to identify the best skill latent vector given a downstream task, which we will utilize in the next phase. More concretely, we mainly choose to employ HILP (Park et al., 2024c) (Section 3) as an unsupervised offline policy pre-training method in our experiments for its state-of-the-art performance in previous benchmarks (Park et al., 2024c). We note, however, that any other unsupervised offline successor feature-based skill learning methods (Touati et al., 2022) or offline goal-conditioned RL methods (Park et al., 2024b) can also be used in place of HILP (see Appendix A.2).

## 4.2 Bridging offline unsupervised RL and online supervised RL

After finishing unsupervised offline policy pre-training, our next step is to convert the learned multi-task skill policy into a task-specific policy that can be fine-tuned to maximize a given downstream reward function $r$ (Figure 1 (bottom middle)). There exist two challenges in this step: (1) we need a mechanism to identify the skill vector $z$ that best solves the given task and (2) we need to reconcile the gap between intrinsic rewards and downstream task rewards for seamless online fine-tuning.

**Skill identification.** Since we chose to use a successor feature- or goal-based unsupervised pre-training method in the previous phase, the first challenge is relatively straightforward. For goal-oriented tasks (*e.g.*, AntMaze (Fu et al., 2020) and Kitchen (Gupta et al., 2020)), we assume the task goal $g$ to be available, and we either directly use $g$ (for goal-conditioned methods) or infer the skill $z^*$ that corresponds to $g$ based on a predefined conversion formula (for successor feature-based methods that support such a conversion (Touati et al., 2022; Park et al., 2024c)). For general reward-maximization tasks, we employ successor feature-based unsupervised pre-training methods, and use the following linear regression to find the skill latent vector $z^*$ that best approximates the downstream task reward function $r : \mathcal{S} \times \mathcal{A} \times \mathcal{S} \to \mathbb{R}$ (Touati et al., 2022; Park et al., 2024c):

$$z^* = \underset{z \in \mathcal{Z}}{\arg\min} \, \mathbb{E}_{(s,a,s') \sim \mathcal{D}_{\texttt{reward}}} \left[ \left( r(s, a, s') - f(s, a, s')^\top z \right)^2 \right], \tag{6}$$

where $f$ is the feature network in the successor feature framework (Section 3) and $\mathcal{D}_{\texttt{reward}}$ is a reward-labeled dataset. This reward-labeled dataset $\mathcal{D}_{\texttt{reward}}$ can be either the full offline dataset $\mathcal{D}_{\texttt{off}}$ (if it is fully reward-labeled), a subset of the offline dataset (if it is partially reward-labeled), or a newly collected dataset with additional environment interactions. In our experiments, we mainly use a small number (*e.g.*, $0.2\%$ for DMC tasks) of reward-labeled samples from the offline dataset for $\mathcal{D}_{\texttt{reward}}$, following previous works (Touati et al., 2022; Park et al., 2024c), but we do not require $\mathcal{D}_{\texttt{reward}}$ to be a subset of $\mathcal{D}_{\texttt{off}}$ (see Appendix A.4).

**Reward scale matching.** After identifying the best skill latent vector $z^*$, our next step is to bridge the gap between intrinsic and extrinsic rewards. Since these two reward functions can have very different scales, naïve online adaptation can lead to abrupt shifts in target Q-values, potentially causing significant performance drops in the early stages of online fine-tuning. While one can employ sophisticated reward-shaping techniques to deal with this issue (Ng et al., 1999; Gleave et al., 2021), in this work, we propose a simple yet effective reward scale matching technique that we find to be effective enough in practice. Specifically, we compute the running mean and standard deviation of intrinsic rewards during the pre-training phase, and normalize the intrinsic rewards with the calculated statistics. Similarly, during the fine-tuning phase, we compute the statistics of task rewards and normalize the task rewards so that they have the same scale and mean as normalized intrinsic rewards. This way, we can prevent abrupt shifts in reward scales without altering the optimal policy for the

| (a) Walker | (b) Cheetah | (c) Quadruped | (d) Jaco | (e) AntMaze-Large | (f) AntMaze-Ultra | (g) Kitchen | (h) Adroit-Pen | (i) Adroit-Door | (j) OGBench-Cube-Single | (k) OGBench-Cube-Double |

Figure 2: **Environments.** We evaluate U2O RL on eleven state-based or pixel-based environments.

downstream task. In our experiments, we find that this simple technique is crucial for achieving good performance, especially in environments with dense rewards (Q6 in Section A.8).

## 4.3 ONLINE FINE-TUNING

Our final step is to fine-tune the skill policy with online environment interactions (Figure 1 (bottom right)). This step is straightforward: since we have found $z^*$ in the previous stage, we can simply *fix* the skill vector $z^*$ in the policy $\pi_\theta(a \mid s, z^*)$ and the Q-function $Q_\phi(s, a, z^*)$, and fine-tune them with the same (offline) RL algorithm used in the first phase (*e.g.*, IQL (Kostrikov et al., 2022), TD3 (Fujimoto et al., 2018)) with additional online interaction data. While one can employ existing specialized techniques for offline-to-online RL for better online adaptation in this phase, we find in our experiments that, thanks to rich representations learned by unsupervised pre-training, simply using the same (offline) RL algorithm is enough to achieve strong performance that matches or even outperforms state-of-the-art offline-to-online RL techniques.

## 4.4 WHY IS U2O RL POTENTIALLY BETTER THAN OFFLINE-TO-ONLINE RL?

Our main claim is that *unsupervised* offline RL is better than supervised offline RL for online fine-tuning. However, this might sound counterintuitive. Especially, if we know the downstream task ahead of time, how can unsupervised offline RL potentially lead to better performance than supervised offline RL, despite the fact that the former does *not* use any task information during the offline phase?

We hypothesize that this is because unsupervised multi-task offline RL enables better *feature learning* than supervised single-task offline RL. By training the agent on a number of diverse intrinsically defined tasks, it gets to acquire rich knowledge about the environment, dynamics, and potential tasks in the form of representations, which helps improve and facilitate the ensuing task-specific online fine-tuning. This resembles the recent observation in machine learning that large-scale unsupervised pre-training improves downstream task performances over task-specific supervised pre-training (Brown et al., 2020; Devlin et al., 2019; Radford et al., 2019; He et al., 2021; 2020; Hénaff et al., 2020). In our experiments, we empirically show that U2O RL indeed learns better features than its supervised counterpart (Q4 in Section 5).

Another benefit of U2O RL is that it does not use any task-specific information during pre-training. This is appealing because we can reuse a single pre-trained policy for a number of different downstream tasks. Moreover, it enables leveraging potentially large, task-agnostic offline data during pre-training, which is often cheaper to collect than task-specific, curated datasets (Lynch et al., 2019).

## 5 EXPERIMENTS

In our experiments, we evaluate the performance of U2O RL in the context of offline-to-online RL. We aim to answer the following research questions: (1) Is U2O RL better than previous offline-to-online RL strategies? (2) Can a single pre-trained model from U2O be fine-tuned to solve multiple tasks? (3) What makes unsupervised offline RL result in better fine-tuning performance than supervised offline RL? (4) Which components of U2O RL are important?

**Environments and offline datasets.** In our experiments, we consider eleven tasks across five benchmarks (Figure 2). **ExORL** (Yarats et al., 2022) is a benchmark suite that consists of offline datasets collected by exploratory policies (*e.g.*, RND (Burda et al., 2019)) on the DeepMind Control Suite (Tassa et al., 2018). We consider four embodiments (Walker, Cheetah, Quadruped, and Jaco), each of which has four tasks. **AntMaze** (Fu et al., 2020; Jiang et al., 2023) is a navigation task, whose goal is to control an 8-DoF quadruped agent to reach a target position. We consider the two most challenging mazes with the largest sizes, `large` and `ultra`, and two types of offline datasets,

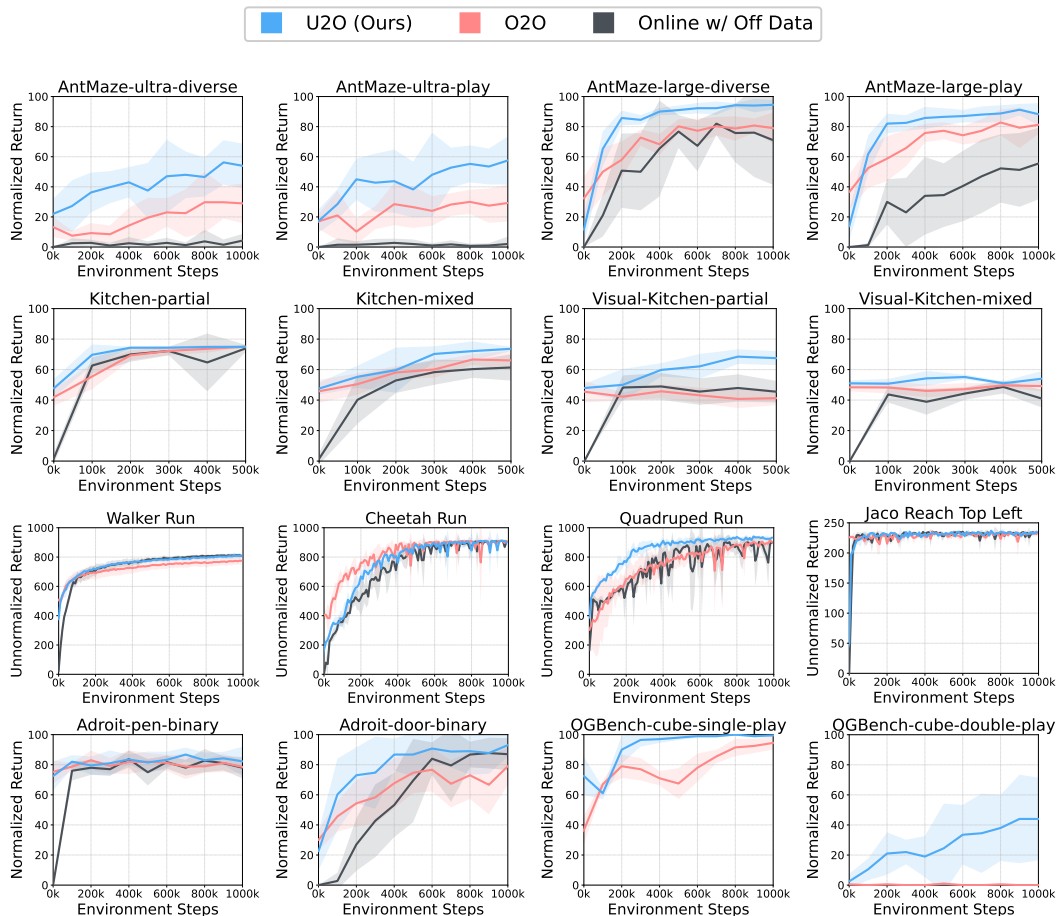

Figure 3: **Online fine-tuning plots of U2O RL and previous offline-to-online RL frameworks (8 seeds).** Across the benchmarks, our U2O RL mostly shows consistently better performance than standard offline-to-online RL and off-policy online RL with offline data.

diverse and play. **Kitchen** (Gupta et al., 2020; Fu et al., 2020) is a robotic manipulation task, where the goal is to control a 9-DoF Franka robot arm to achieve four subtasks sequentially. We consider two types of offline datasets from the D4RL suite (Fu et al., 2020), partial and mixed. **Visual Kitchen** (Gupta et al., 2020; Fu et al., 2020; Park et al., 2024c) is a pixel-based variant of the Kitchen environment, where an agent must achieve four subtasks purely from $64 \times 64 \times 3$ pixel observations instead of low-dimensional state information. **Adroit** (Fu et al., 2020) is a dexterous manipulation benchmark, where the goal is to control a 24-DoF robot hand to twirl a pen or open a door. **OGBench-Cube** (Park et al., 2024a) is an additional manipulation benchmark whose goal is to control a 6-DoF UR5e robot arm to perform pick-and-place manipulation of multiple cubes from an unlabeled, diverse dataset. We use a single-task version of OGBench-Cube to make it compatible with our offline-to-online RL setting. We provide further details in Appendix C.1.

**Implementation.** In our experiments, we mainly employ HILP (Park et al., 2024c) as the unsupervised offline policy pre-training algorithm in U2O RL. For the offline RL backbone, we use TD3 (Fujimoto et al., 2018) for ExORL and IQL (Kostrikov et al., 2021) for others following previous works (Touati et al., 2022; Park et al., 2024c). Since both IQL and TD3+BC (Fujimoto & Gu, 2021; Tarasov et al., 2023a) have been known to achieve strong performance in the offline-to-online RL setting (Tarasov et al., 2023b), we use them for the online fine-tuning phase in U2O RL as well. For sparse-reward tasks (AntMaze, Kitchen, and Adroit), we do not apply reward scale matching. For AntMaze, Kitchen, and Adroit, we report normalized scores, following Fu et al. (2020). In our experiments, we use **8 random seeds** and report standard deviations with shaded areas, unless otherwise stated. We refer the reader to Appendix C for the full implementation details and hyperparameters.

**Q1. Is U2O RL better than previous offline-to-online RL frameworks?**

Table 1: **Comparison between U2O RL and previous offline-to-online RL methods.** We denote how performances change before and after online fine-tuning with arrows. Baseline scores except RLPD (Ball et al., 2023) are taken from Nakamoto et al. (2023); Wang et al. (2023a). Scores within the 5% of the best score are highlighted in bold, as in Kostrikov et al. (2022). We use 8 random seeds for each task for U2O RL.

| Task | antmaze-ultra-diverse | antmaze-ultra-play | antmaze-large-diverse | antmaze-large-play | kitchen-partial | kitchen-mixed |
|---|---|---|---|---|---|---|
| CQL | - | - | $25 \rightarrow 87$ | $34 \rightarrow 76$ | $71 \rightarrow \mathbf{75}$ | $56 \rightarrow 50$ |
| IQL | $13 \rightarrow 29$ | $17 \rightarrow 29$ | $40 \rightarrow 59$ | $41 \rightarrow 51$ | $40 \rightarrow 60$ | $48 \rightarrow 48$ |
| AWAC | - | - | $00 \rightarrow 00$ | $00 \rightarrow 00$ | $01 \rightarrow 13$ | $02 \rightarrow 12$ |
| O3F | - | - | $59 \rightarrow 28$ | $68 \rightarrow 01$ | $11 \rightarrow 22$ | $06 \rightarrow 33$ |
| ODT | - | - | $00 \rightarrow 01$ | $00 \rightarrow 00$ | - | - |
| CQL+SAC | - | - | $36 \rightarrow 00$ | $21 \rightarrow 00$ | $71 \rightarrow 00$ | $59 \rightarrow 01$ |
| Hybrid RL | - | - | $\rightarrow 00$ | $\rightarrow 00$ | $\rightarrow 00$ | $\rightarrow 01$ |
| SAC+od | - | - | $\rightarrow 00$ | $\rightarrow 00$ | $\rightarrow 07$ | $\rightarrow 00$ |
| SAC | - | - | $\rightarrow 00$ | $\rightarrow 00$ | $\rightarrow 03$ | $\rightarrow 02$ |
| IQL+od | $\rightarrow 04$ | $\rightarrow 05$ | $\rightarrow 71$ | $\rightarrow 56$ | $\rightarrow 74$ | $\rightarrow 61$ |
| FamO2O | - | - | $\rightarrow 64$ | $\rightarrow 61$ | - | - |
| RLPD | $00 \rightarrow 00$ | $00 \rightarrow 00$ | $00 \rightarrow \mathbf{94}$ | $00 \rightarrow 81$ | - | - |
| Cal-QL | $05 \rightarrow 05$ | $15 \rightarrow 13$ | $33 \rightarrow \mathbf{95}$ | $26 \rightarrow \mathbf{90}$ | $67 \rightarrow \mathbf{79}$ | $38 \rightarrow \mathbf{80}$ |
| **U2O (Ours)** | $22 \rightarrow \mathbf{54}$ | $17 \rightarrow \mathbf{58}$ | $11 \rightarrow \mathbf{95}$ | $14 \rightarrow \mathbf{88}$ | $48 \rightarrow 75$ | $48 \rightarrow 74$ |

We begin our experiments by comparing our approach, unsupervised-to-online RL, with two previous offline-to-online RL *frameworks* (Section 2): **offline-to-online RL** (**O2O RL**) and **off-policy online RL with offline data** (**Online w/ Off Data**). To recall, offline-to-online RL (Lee et al., 2022; Nair et al., 2020; Nakamoto et al., 2023; Yu & Zhang, 2023; Lei et al., 2023) first pre-trains a policy with *supervised* offline RL using the task reward, and then continues training it with online rollouts. Off-policy online RL (Ball et al., 2023; Luo et al., 2024; Song et al., 2023) trains a policy from scratch on top of a replay buffer filled with offline data. Here, we use the *same* offline RL backbone (*i.e.*, TD3 for ExORL and IQL for AntMaze, Kitchen, and Adroit) to ensure apples-to-apples comparisons between the three frameworks. We will compare U2O RL with previous specialized offline-to-online RL techniques in Q2 of Section 5.

Figure 3 shows the online fine-tuning curves on 14 different tasks. The results suggest that U2O RL generally leads to better performance than both offline-to-online RL and off-policy online RL across the environments, despite not using any task information during pre-training. Notably, U2O RL significantly outperforms these two previous frameworks in the most challenging AntMaze tasks (`antmaze-ultra-{diverse, play}`).

**Q2. How does U2O RL compare to previous specialized offline-to-online RL techniques?**

Next, we compare U2O RL with 13 previous specialized offline-to-online RL methods, including **CQL** (Kumar et al., 2020), **IQL** (Kostrikov et al., 2022), **AWAC** (Nair et al., 2020), **O3F** (Mark et al., 2022), **ODT** (Zheng et al., 2022), **CQL+SAC** (Kumar et al., 2020; Haarnoja et al., 2018), **Hybrid RL** (Song et al., 2023), **SAC+od (offline data)** (Haarnoja et al., 2018; Ball et al., 2023), **SAC** (Haarnoja et al., 2018), **IQL+od (offline data)** (Kostrikov et al., 2022; Ball et al., 2023), **FamO2O** (Wang et al., 2023a), **RLPD** (Ball et al., 2023), and **Cal-QL** (Nakamoto et al., 2023). We show the comparison results in Table 1, where we take the scores from Nakamoto et al. (2023); Wang et al. (2023a) for the tasks that are common to ours. Since Cal-QL achieves the best performance in the table, we additionally make a comparison with Cal-QL on `antmaze-ultra-{diverse, play}` as well, by running their official implementation with tuned hyperparameters.

Table 1 shows that U2O RL achieves strong performance that matches or sometimes even outperforms previous offline-to-online RL methods, even though U2O RL does *not* use any task information during offline pre-training nor any specialized offline-to-online techniques. In particular, in the most challenging `antmaze-ultra` tasks, U2O RL outperforms the previous best method (Cal-QL) by a significant margin. This is very promising because, even if U2O RL does not necessarily outperform the state-of-the-art methods on every single task (though it is at least on par with the previous best methods), U2O RL enables reusing a single unsupervised pre-trained policy for multiple downstream tasks, unlike previous offline-to-online RL methods that perform *task-specific* pretraining.

**Q3. Can a single pre-trained model from U2O be fine-tuned to solve multiple tasks?**

One important advantage of U2O RL is that it can reuse a single task-agnostic dataset for multiple different downstream tasks, unlike standard offline-to-online RL. To demonstrate this, we train U2O RL with four different tasks from the same task-agnostic ExORL dataset on each DMC environment, and report the full training curves in Figure 7 of Appendix A.1. The results show that, for example,

a single pre-trained model on the Walker domain can be fine-tuned for all four tasks (Walker Run, Walker Flip, Walker Stand, and Walker Walk). Note that even though U2O RL uses a single task-agnostic pre-trained model, the performance of U2O RL matches or even outperforms O2O RL, which pre-trains a model with task-specific rewards.

**Q4. Why does U2O RL often outperform supervised offline-to-online RL?**

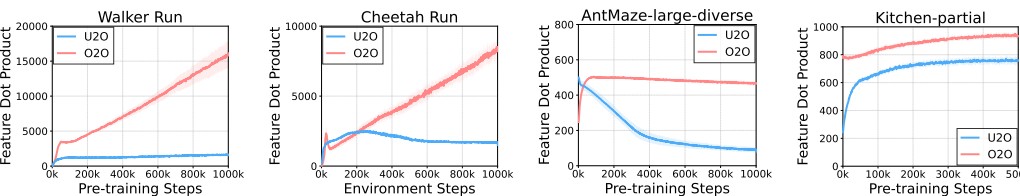

Figure 4: **Feature dot products during offline RL pre-training (lower is better, 8 seeds).** The plots show that *unsupervised* offline pre-training effectively prevents feature collapse (co-adaptation), yielding better representations than supervised offline pre-training.

In the above experiments, we showed that U2O RL often even outperforms previous supervised offline-to-online RL methods. We hypothesized in Section 4.4 that this is because unsupervised offline pre-training yields better *representations* that facilitate online task adaptation. To empirically verify this hypothesis, we measure the quality of the value function representations using the method proposed by Kumar et al. (2022). Specifically, we define the value features $\zeta_\phi(s, a)$ as the penultimate layer of the value function $Q_\phi$, *i.e.*, $Q_\phi(s, a) = w_\phi^\top \zeta_\phi(s, a)$, and measure the dot product between consecutive state-action pairs, $\zeta_\phi(s, a)^\top \zeta_\phi(s', a')$ (Kumar et al., 2022). Intuitively, this dot product represents the degree to which these two representations are "collapsed" (or "co-adapted"), which is known to be correlated with poor performance (Kumar et al., 2022) (*i.e.*, the lower the better).

Figure 4 compares the dot product metrics of unsupervised offline RL (in U2O RL) and supervised offline RL (in O2O RL) on four benchmark tasks. The results suggest that our unsupervised multi-task pre-training effectively prevents feature co-adaptation and thus indeed yields better representations across the environments. This highlights the benefits of unsupervised offline pre-training, and (partially) explains the strong online fine-tuning performance of U2O RL. We additionally provide further analyses with different offline unsupervised RL algorithms (*e.g.*, graph Laplacian-based successor feature learning (Touati et al., 2022; Wu et al., 2019b)) in Appendix A.2.

**Q5. Is fine-tuning better than other alternative strategies (*e.g.*, hierarchical RL)?**

In this work, we focus on the *fine-tuning* of offline pre-trained skill policies, but this is not the only way to leverage pre-trained skills for downstream tasks. To see how our fine-tuning scheme compares to other alternative strategies, we compare U2O RL with three previously considered approaches: **hierarchical RL** (**HRL**, *e.g.*, **OPAL** (Ajay et al., 2021), **SPiRL** (Pertsch et al., 2021)) (Ajay et al., 2021; Pertsch et al., 2021; Touati et al., 2022; Park et al., 2024c; Hu et al., 2023), **zero-shot RL** (Touati et al., 2022; Park et al., 2024c), and **PEX** (Zhang et al., 2023). HRL additionally trains a high-level policy that

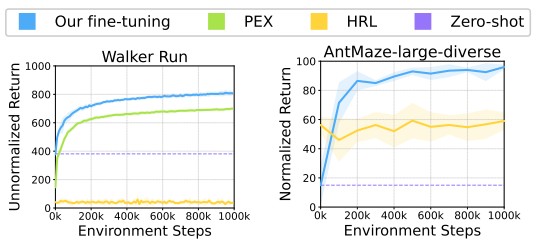

Figure 5: **Fine-tuning is better than previous strategies, such as hierarchical RL, zero-shot RL, and PEX (8 seeds).**

combines fixed pre-trained skills in a sequential manner. Zero-shot RL simply finds the skill policy that best solves the downstream task, with no fine-tuning or hierarchies. PEX combines fixed pre-trained multi-task policies and a newly initialized policy with a multiplexer that chooses the best policy.

Figure 5 shows the comparison results on top of the same pre-trained unsupervised skill policy. Since PEX is not directly compatible with IQL, we evaluate PEX only on the tasks with TD3 (*e.g.*, ExORL tasks). The plots suggest that our fine-tuning strategy leads to significantly better performance than previous approaches. This is because pre-trained offline skill policies are often not perfect (due to the limited coverage or suboptimality of the dataset), and thus using a fixed offline policy is often

not sufficient to achieve strong performance in downstream tasks. We provide additional results in Appendix A.6.

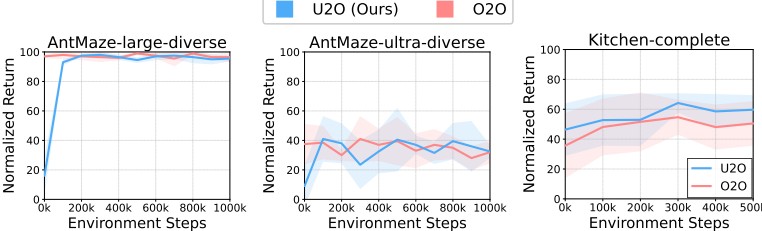

Figure 6: **Online RL learning curves with expert-only datasets (4 seeds).**

**Q6. Negative results: When is U2O RL better than O2O RL?**

While we showed strong results of U2O RL throughout the paper, U2O RL is not *always* better than O2O RL. Specifically, U2O RL may not be as effective when the dataset is monolithic (*e.g.*, consists of only expert trajectories, has less diversity, etc.). To empirically show this, we conduct an additional experiment with a different dataset in `antmaze-large` and `antmaze-ultra` that consists of monolithic, expert trajectories (we collect a 1M-sized dataset by rolling out an offline pre-trained policy) as well as `kitchen-complete` dataset, which also consists of expert trajectories. Figure 6 shows the (negative) results, which suggest that U2O RL is not particularly better than O2O RL on these monolithic, optimal datasets. However, we emphasize that U2O RL achieves similar final performance to O2O RL even on these datasets, and has the unique strength that a single unsupervised pre-trained model can be fine-tuned to many different reward functions, unlike standard O2O RL.

We refer to the reader to Appendix for further analysis including (1) combining U2O RL with other offline unsupervised skill learning methods (Appendix A.2), (2) comparisons between U2O RL and pure representation learning schemes (Appendix A.3), (3) U2O RL without reward samples in the bridging phase (Appendix A.4), (4) an ablation study with different skill identification strategies (Appendix A.5), (5) additional results with different online RL strategies (Appendix A.6), (6) comparison to O2O RL combined with methods for mitigating the feature collapse issue (Appendix A.7), and (7) ablation studies on each component in U2O RL such as reward scale matching, value transfer and policy transfer (Appendix A.8).

## 6 CONCLUSION

In this work, we investigated how unsupervised pre-training of diverse policies enables better online fine-tuning than standard supervised offline-to-online RL. We showed that our unsupervised-to-online recipe often achieves even better performance and stability than previous offline-to-online RL approaches, thanks to the rich representations learned by pre-training on diverse tasks. We also demonstrated that U2O RL enables reusing a single offline pre-trained policy for multiple downstream tasks.

**Limitation.** As shown in Q7 of Section 5, U2O RL is not necessarily better than O2O RL when the offline dataset is monolithic and heavily tailored toward the downstream task. We believe U2O RL is most effective (compared to standard offline-to-online RL) when the dataset is highly diverse so that the unsupervised offline RL method can learn a variety of behaviors and thus learn better features and representations. Given the recent successes in large-scale self-supervised and unsupervised pre-training from unlabeled data, we believe U2O RL serves as a step toward a general recipe for scalable data-driven decision-making.

## REPRODUCIBILITY STATEMENT

We provide implementation details in Section 5 and Appendix C including hyperparameters. We also provide pseudo-code in Appendix B and have attached our source code to the OpenReview submission page.

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

# A ADDITIONAL EXPERIMENTS

## A.1 FULL EXORL EXPERIMENTS

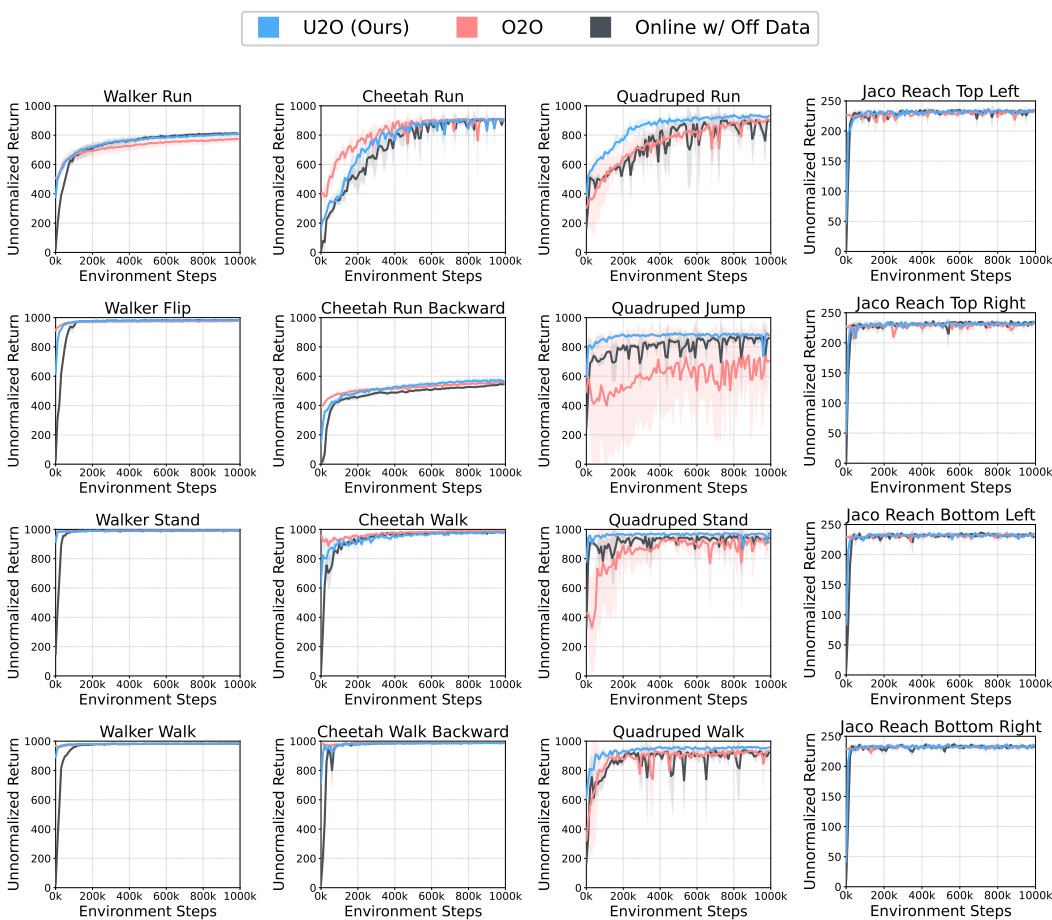

Figure 7: **Learning curves during online RL fine-tuning (8 seeds).** A single pre-trained model from U2O can be fine-tuned to solve multiple downstream tasks. Across the embodiments and tasks, our U2O RL matches or outperforms standard offline-to-online RL and off-policy online RL with offline data even though U2O RL uses a single task-agnostic pre-trained model.

## A.2 CAN U2O RL BE COMBINED WITH OTHER OFFLINE UNSUPERVISED RL METHODS?

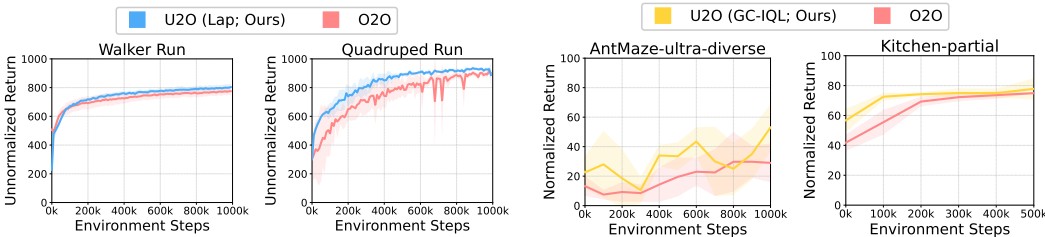

Figure 8: **U2O RL with Laplacian-based successor feature learning (8 seeds).**

Figure 9: **U2O RL with goal-conditioned IQL (8 seeds).**

While we employ HILP (Park et al., 2024c) as an offline unsupervised skill learning method in U2O RL in our main experiments, our recipe can be combined with other offline unsupervised skill

learning methods as well. To show this, we replace HILP with a graph Laplacian-based successor feature method (Touati et al., 2022; Wu et al., 2019b) or goal-conditioned IQL (GC-IQL) (Kostrikov et al., 2022; Park et al., 2024b), and report the results in Figures 8 and 9, respectively. The results demonstrate that U2O RL with different unsupervised RL methods also improves performance over standard offline-to-online RL.

Additionally, we show that other unsupervised skill learning methods also lead to better value representations. We measure the same feature dot product metric in Section 5 with the graph Laplacian-based successor feature learning method and report the results in Figure 10. The results suggest that this unsupervised RL method also prevents feature co-adaptation, leading to better features.

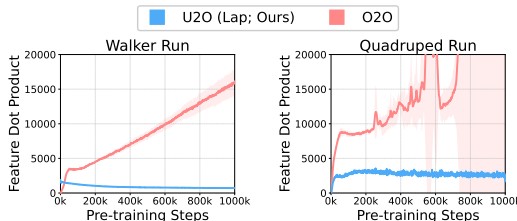

Figure 10: **Feature dot product analysis with Laplacian-based successor feature learning (8 seeds).**

### A.3  DO WE NEED TO USE UNSUPERVISED RL FOR PRE-TRAINING REPRESENTATIONS?

In Sections 4.4 and 5, we hypothesized and empirically showed that U2O RL is often better than O2O RL because it learns better representations. This leads to the following natural question: do we *need* to use offline unsupervised *reinforcement learning*, as opposed to general representation learning? To answer this question, we consider two pure representation learn-

Table 2: **Comparison between U2O RL and pure representation learning algorithms (4 seeds).**

| Task | antmaze-large-diverse |
|---|---|
| U2O (HILP, Q Ours) | **94.50 ± 3.16** |
| U2O (HILP, $\xi$) | 5.50 ± 1.91 |
| Temporal contrastive learning | 37.50 ± 15.00 |

ing algorithms as alternatives to unsupervised RL: temporal contrastive learning (Eysenbach et al., 2022) and Hilbert (metric) representation learning (Park et al., 2024c), where the latter is equivalent to directly taking $\xi$ in the HILP framework (Equation 4) (note that the original U2O RL takes the Q function of HILP, not the Hilbert representation $\xi$ itself, which is used to train the Q function). To evaluate their fine-tuning performances, for the temporal contrastive representation, we fine-tune both the Q function and policy with contrastive RL (Eysenbach et al., 2022); for the Hilbert representation, we take the pre-trained representation, add one new layer, and use it as the initialization of the Q function. Table 2 shows the results on `antmaze-large-diverse`. Somewhat intriguingly, the results suggest that it is important to use the full unsupervised RL procedure, and pure representation learning methods result in much worse performance in this case. This becomes more evident if we compare U2O RL (HILP Q, ours) and U2O RL (HILP $\xi$), given that they are rooted in the same Hilbert representation. We believe this is because, if we simply use an off-the-shelf representation learning, there exists a discrepancy in training objectives between pre-training (*e.g.*, metric learning) and fine-tuning (Q-learning). On the other hand, in U2O RL, we pre-train a representation with unsupervised Q-learning (though with a different reward function), and thus the discrepancy between pre-training and fine-tuning becomes less severe.

### A.4  CAN WE DO "BRIDGING" WITHOUT ANY REWARD-LABELED DATA?

In the bridging phase of U2O RL (Section 4.2), we assume a (small) reward-labeled dataset $\mathcal{D}_{\texttt{reward}}$. In our experiments, we sample a small number of transitions (*e.g.*, 0.2% in the case of DMC) from the offline dataset and label them with the ground-truth reward function, as in prior works (Touati et al., 2022; Park et al., 2024c). However, these samples do not necessarily have to come from the offline dataset. To show this, we conduct an additional experiment where we do not assume access to any of the existing re-

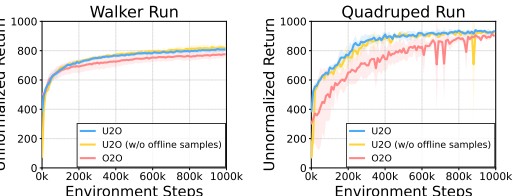

Figure 11: **U2O RL without using reward-labeling in the offline dataset (8 seeds).**

ward samples or the ground-truth reward function in the bridging phase. Specifically, we collect 10K

online samples with random skills and perform the linear regression in Equation 6 only using the collected online transitions. We report the performances of U2O (without offline samples) and O2O in Figure 11. The results show that U2O still works and outperforms the supervised offline-to-online RL baseline.

## A.5 HOW DO DIFFERENT STRATEGIES OF SKILL IDENTIFICATION AFFECT PERFORMANCE?

To understand how skill identification strategies affect online RL performance, we compare our strategy in Section 4.2 with an alternative strategy that simply selects a random latent vector $z$ from the skill space. Figure 12 shows that the skill identification with a randomly selected latent vector performs worse than our strategy. This is likely because modulating the policy with the best latent vector helps boost task-relevant exploration and information.

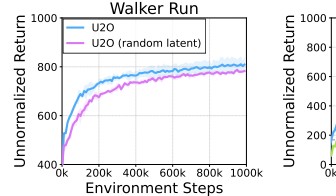 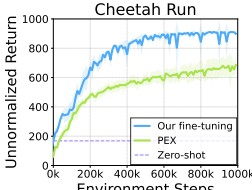

Figure 12: **Ablation study of skill identification (4 seeds).**

Figure 13: **Comparison with PEX and zero-shot RL (4 seeds).**

## A.6 ADDITIONAL EXPERIMENTS ON FINE-TUNING STRATEGIES

We additionally provide experimental results of fine-tuning strategies on a different task (*i.e.*, Cheetah Run). Figure 13 shows that our fine-tuning strategy outperforms previous strategies, such as zero-shot RL and PEX. This result further supports the effectiveness of fine-tuning.

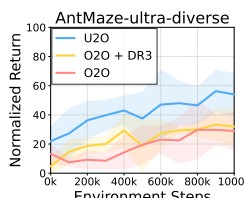

Figure 14: **Comparison with O2O RL + DR3 (4 seeds).**

## A.7 HOW DOES U2O PERFORM COMPARED TO O2O COMBINED WITH METHODS FOR MITIGATING FEATURE COLLAPSE?

To further understand the effectiveness of U2O RL, we compare the performance of U2O RL with that of O2O RL combined with DR3 (Kumar et al., 2022), a regularizer that regularizes feature dot products to prevent the feature collapse issue. The result in Figure 14 shows that simply adding the DR3 regularizer is not as effective as U2O RL. We believe this is likely because the full unsupervised RL procedure can lead to much richer representations than simply adding a regularizer.

## A.8 HOW DOES EACH COMPONENT IN U2O RL AFFECT PERFORMANCE?

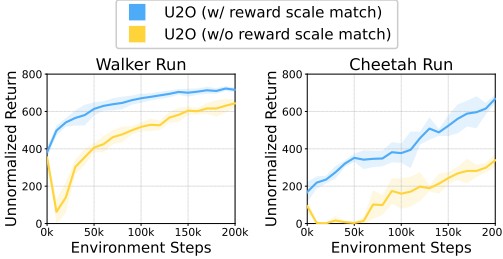 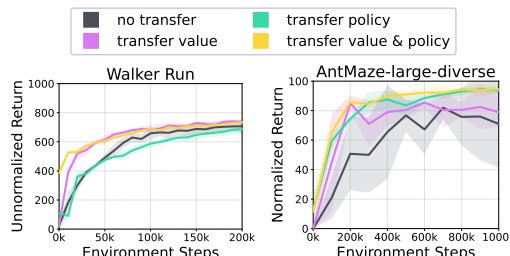

Figure 15: **Ablation study of reward scale matching (4 seeds).**

Figure 16: **Ablation study of value transfer and policy transfer (4 seeds).**

**Reward scale matching.** In Section 4.2, we propose a simple reward scale matching technique that bridges the gap between intrinsic rewards and downstream task rewards. We ablate this component, and report the results in Figure 15. The results suggest that our reward scale matching technique effectively prevents a performance drop at the beginning of the online fine-tuning stage, leading to substantially better final performance on dense-reward tasks (*e.g.*, Walker Run and Cheetah Run).

**Value transfer vs. policy transfer.** In U2O RL, we transfer *both* the value function and policy from unsupervised pre-training to supervised fine-tuning. To dissect the importance of each component, we conduct an ablation study, in which we compare four settings: (1) without any transfer, (2) value-only transfer, (3) policy-only transfer, and (4) full transfer. Figure 16 demonstrates the ablation results on Walker and AntMaze. The results suggest that both value transfer and policy transfer matter in general, but value transfer is more important than policy transfer. This aligns with our findings in Q4 of Section 5 as well as Kumar et al. (2022), which says that the quality of value features often correlates with the performance of TD-based RL algorithms.

## B   ALGORITHM TABLE

---

**Algorithm 1** U2O RL: Unsupervised-to-Online Reinforcement Learning

---

**Require**: offline dataset $\mathcal{D}_{\texttt{off}}$, reward-labeled dataset $\mathcal{D}_{\texttt{reward}}$, empty replay buffer $\mathcal{D}_{\texttt{on}}$, offline pre-training steps $N_{\texttt{PT}}$, online fine-tuning steps $N_{\texttt{FT}}$, skill latent space $\mathcal{Z}$
Initialize the parameters of policy $\pi_\theta$ and action-value function $Q_\phi$
**for** $t = 0, 1, 2, \ldots N_{\texttt{PT}} - 1$ **do**
    Sample transitions $(s, a, s')$ from $\mathcal{D}_{\texttt{off}}$
    Sample latent vector $z \in \mathcal{Z}$ and compute intrinsic rewards $r^{\texttt{int}}$
    Update policy $\pi_\theta(a \mid s, z)$ and $Q_\phi(s, a, z)$ using normalized intrinsic rewards $\tilde{r}^{\texttt{int}}$
**end for**
Compute the best latent vector $z^*$ with Equation 6 using samples $(s, a, s', r)$ from $\mathcal{D}_{\texttt{reward}}$
**for** $t = 0, 1, 2, \ldots N_{\texttt{FT}} - 1$ **do**
    Collect transition $(s, a, s', r)$ via environment interaction with $\pi_\theta$ and add to replay buffer $\mathcal{D}_{\texttt{on}}$
    Sample transitions $(s, a, s', r)$ from $\mathcal{D}_{\texttt{off}} \cup \mathcal{D}_{\texttt{on}}$
    Update policy $\pi_\theta(a \mid s, z^*)$ and $Q_\phi(s, a, z^*)$ using normalized task rewards $\tilde{r}$
**end for**

---

## C   EXPERIMENTAL DETAILS

For offline RL pre-training, we use 1M training steps for ExORL, AntMaze, and Adroit and 500K steps for Kitchen, following Park et al. (2024c). For online fine-tuning, we use 1M additional environment steps for ExORL, AntMaze, and Adroit and 500K steps for Kitchen with an update-to-data ratio of 1. We implement U2O RL based on the official implementation of HILP (Park et al., 2024c). We evaluate the normalized return with 50 episodes every 10k online steps for ExORL tasks, and every 100k online steps for AntMaze, Kitchen, and Adroit tasks. We run our experiments on A5000 or RTX 3090 GPUs. Each run takes at most 40 hours (*e.g.* Visual Kitchen). We provide our implementation in the supplementary material.

### C.1   ENVIRONMENTS AND DATASETS

**ExORL (Yarats et al., 2022).** In the ExORL benchmark, we consider four embodiments, Walker, Cheetah, Quadruped, and Jaco. Each embodiment has four tasks: Walker has {Run, Flip, Stand, Walk}, Cheetah has {Run, Run Backward, Walk, Walk Backward}, Quadruped has {Run, Jump, Stand, Walk}, and Jaco has {Reach Top Left, Reach Top Right, Reach Bottom Left, Reach Bottom Right}. For all the tasks in Walker, Cheetah, and Quadruped, the maximum return is 1000, and Jaco has 250. Each embodiment has an offline dataset, which is collected by running exploratory agents such as RND (Burda et al., 2019), and then annotated with task reward function. We use the first 5M transitions of the offline dataset following the prior work (Touati et al., 2022; Park et al., 2024c). The maximum episode length is 250 (Jaco) or 1000 (others).

**AntMaze (Fu et al., 2020; Jiang et al., 2023).** In AntMaze, a quadruped agent aims at reaching the (pre-defined) target position in a maze and gets a positive reward when the agent arrives at a pre-defined neighborhood of the target position. We consider two types of Maze: `antmaze-large` (Fu et al., 2020) and `antmaze-ultra` (Jiang et al., 2023), where the latter has twice the size of the former. Each maze has two types of offline datasets: `play` and `diverse`. The dataset consists of 999 trajectories with an episode length of 1000. In each trajectory, an agent is initialized at a

random location in the maze and is directed to an arbitrary location, which may not be the same as the target goal. At the evaluation, `antmaze-large` has a maximum episode length of 1000, and `antmaze-ultra` has 2000. We report normalized scores by multiplying the returns by 100.

**Kitchen (Gupta et al., 2020; Fu et al., 2020).** In the Kitchen environment, a Franka robot should achieve four sub-tasks, `microwave`, `slide cabinet`, `light switch`, and `kettle`. Each task has a success criterion determined by an object configuration. Whenever the agent achieves a sub-task, a task reward of 1 is given, where the maximum return is 4. We consider two types of offline datasets: `mixed` and `partial`. We report normalized scores by multiplying the returns by 100. For Visual-Kitchen, we follow the same camera configuration as Mendonca et al. (2021), Park et al. (2024d), and Park et al. (2024c), to render $64 \times 64$ RGB observations, which are used instead of low-dimensional states. We report normalized scores by multiplying the returns by 25.

**Adroit (Fu et al., 2020).** In Adroit, a 24-DoF Shadow Hand robot should be controlled to achieve a desired task. We consider two tasks: `pen-binary` and `door-binary`, following prior works (Ball et al., 2023; Li et al., 2023). The maximum episode lengths of `pen-binary` and `door-binary` are 100 and 200. respectively. We report normalized scores by multiplying the returns by 100.

**OGBench-Cube (Park et al., 2024a).** In the OGBench-Cube, a 6-DoF UR5e robot arm should be controlled to arrange multiple cubes into the desired configuration. The maximum episode lengths for `cube-single` and `cube-double` are 200 and 500, respectively. To make the originally goal-conditioned tasks compatible with regular (single-task) offline-to-online RL, we fix the `task_id` to 2, and define the task reward as the negative of the number of the unmatched cubes. We report binary success rates multiplied by 100.

## C.2 HYPERPARAMETERS

Table 3: Hyperparameters of unsupervised RL pre-training in ExORL.

| Hyperparameter | Value |
|---|---|
| Learning rate | 0.0005 (feature), 0.0001 (others) |
| Optimizer | Adam (Kingma & Ba, 2015) |
| Minibatch size | 1024 |
| Feature MLP dimensions | $(512, 512)$ |
| Value MLP dimensions | $(1024, 1024, 1024)$ |
| Policy MLP dimensions | $(1024, 1024, 1024)$ |
| TD3 target smoothing coefficient | 0.01 |
| TD3 discount factor $\gamma$ | 0.98 |
| Latent dimension | 50 |
| State samples for latent vector inference | 10000 |
| Successor feature loss | Q loss |
| Hilbert representation discount factor | 0.96 (Walker), 0.98 (others) |
| Hilbert representation expectile | 0.5 |
| Hilbert representation target smoothing coefficient | 0.005 |

Table 4: Hyperparameters of unsupervised RL pre-training in AntMaze, Kitchen, and Adroit.

| Hyperparameter | Value |
|---|---|
| Learning rate | 0.0003 |
| Optimizer | Adam (Kingma & Ba, 2015) |
| Minibatch size | 256 (Adroit), 512 (others) |
| Value MLP dimensions | $(256, 256, 256)$ (Adroit), $(512, 512, 512)$ (others) |
| Policy MLP dimensions | $(256, 256, 256)$ (Adroit), $(512, 512, 512)$ (others) |
| Target smoothing coefficient | 0.005 |
| Discount factor $\gamma$ | 0.99 |
| Latent dimension | 32 |
| Hilbert representation discount factor | 0.99 |
| Hilbert representation expectile | 0.95 |
| Hilbert representation target smoothing coefficient | 0.005 |
| HILP IQL expectile | 0.9 (AntMaze), 0.7 (others) |
| HILP AWR temperature | 0.5 (Kitchen) 3 (Adroit-door), 10 (others) |

