# OpenReview forum: "Unsupervised-to-Online Reinforcement Learning"
_ICLR.cc/2025/Conference — Submitted to ICLR 2025_

### Official Review · Reviewer_JtA5 · 2024-10-20

**Soundness:** 4
**Presentation:** 4
**Contribution:** 3
**Rating:** 6
**Confidence:** 3

**Summary:**

This paper introduces a novel concept termed unsupervised-to-online reinforcement learning (U2O), which replaces the conventional task-specific offline RL pretraining in the offline2online RL paradigm with task-agnostic unsupervised offline RL pretraining. The study reveals that unsupervised pretraining enables agents to adapt more quickly to downstream tasks compared to domain-specific offline RL pretraining. This advantage stems from the richer and more general-purpose representations fostered by unsupervised learning, which encode a broader range of task features and thus prevent feature collapse specific to any single task. The paper presents a practical and straightforward implementation of their U2O method. Empirical experiments conducted on state-based and image-based tasks, along with extensive ablations, demonstrate the effectiveness of leveraging unsupervised pretraining for general-purpose representation over traditional offline RL pretraining.

**Strengths:**

1. Replacing O2O with U2O is a reasonable and innovative approach.
2. The authors comprehensively discuss the advantages of U2O over O2O, supported by extensive empirical evidence.
3. Experimental results indicate that U2O outperforms O2O in terms of performance.
4. The paper is well-written, clearly highlighting the challenges and contributions.
5. The detailed explanations in the Experiments and Appendix effectively justify why U2O is superior to O2O.

**Weaknesses:**

1. One potential limitation is the skill identification process. In this paper, the authors determine the optimal skill $z*$ by minimizing the MSE loss w.r.t to the single-step reward using a small reward-labeled dataset. Ideally, a skill should encapsulate more long-horizon information and should not be solely defined by single-step rewards. More advanced skill identification process can be developed in the future.
2. The scope of this paper is relatively narrow. In my view, unsupervised pretraining can facilitate various types of fine-tuning, including offline RL fine-tuning, imitation learning, and others. Limiting the study to only online RL fine-tuning could be expanded for a more comprehensive evaluation.

**Questions:**

N/A

---

> ### Author Response · Authors · 2024-11-20
> **Rebuttal by Authors**
>
> Thank you for the detailed review. We appreciate the reviewer’s questions about the skill selection strategy as well as alternative ways to adapt unsupervised RL policies. Please find our response below. The revisions made are marked with “$\text{\color{red}red}$” in the revised paper.
>
>
>
> **Q1. More advanced skill identification processes can be developed for encapsulating long-horizon information**
>
> **A1.** Thanks for the question. While we found in our experiments that fixing a single $z$ was sufficient for the various tasks we used, we agree that U2O may benefit from a more sophisticated skill identification strategy for even longer-horizon and more complex tasks. We leave this extension for an interesting future research direction.
>
>
> **Q2. "Unsupervised pretraining can facilitate various types of fine-tuning, including offline RL fine-tuning, imitation learning, and others. Limiting the study to only online RL fine-tuning could be expanded for a more comprehensive evaluation."**
>
> **A2.** Thanks for the suggestion! In Q4 of Section 5, we tested various strategies to utilize a pre-trained unsupervised RL policy — i.e., (1) online RL fine-tuning, (2) zero-shot RL, (3) hierarchical RL, and (4) PEX, and found online RL fine-tuning to be the best among the four on our tasks. However, as the reviewer pointed out, there might be even more diverse ways and different problem settings (e.g., imitation learning) to adapt an unsupervised RL policy. We leave investigating such a possibility as future work.
>
> We thank the reviewer again for providing valuable comments and feedback. We hope that our response has addressed the questions raised in the review, and please feel free to let us know if there are any additional concerns or questions.

---

> ### Comment · Reviewer_JtA5 · 2024-11-27
>
> Thanks for the detailed responses.
>
> \+ Although most reviewers believe this paper is a naive combination of existing methods, I think the contribution of proposing the U2O framework is significant, since it holds the potential to utilize a broader range of task-agnostic data to faciliate efficient task-specific training, which is especially crucial when deploying RL in many applications where task-specific data is limited.
>
> \- I'm curious about the impact on online RL fine-tuning if offline RL pretraining incorporates regularizations like DR3 to prevent feature collapse. It seems the key advantage of U2O over O2O doesn't stem from this factor.
>
> \-  I agree with other reviewers that the paper may overstate some claims.
>
> Considering the potential advantages of leveraging more task-agnositc data, I still maintain a positive score. But, due to the remaining unresolved concerns, I will maintain my score to a boardline accept.

---

> > ### Author Response · Authors · 2024-12-01
> > **Response to Reviewer JtA5**
> >
> > Thanks again for your comments. We respond to your follow-up comments as follows.
> >
> > **Q3. I'm curious about the impact on online RL fine-tuning if offline RL pretraining incorporates regularizations like DR3 to prevent feature collapse. It seems the key advantage of U2O over O2O doesn't stem from this factor.**
> >
> > **A3.** Thanks for the question. Following the suggestions from the reviewers, we have added additional results (and will add more results; please see **A10** for Reviewer rCe7) with the DR3 regularizer, showing that simply adding a regularizer is not sufficient to match the performance from full unsupervised pre-training.
> >
> > **Q4.** I agree with other reviewers that the paper may overstate some claims.
> >
> > **A4.** Thanks for the feedback. Following the suggestions, we have revised the paper to replace “framework” with “recipe”, and have clarified that we assume a skill-based unsupervised offline RL algorithm throughout the paper.

---

### Official Review · Reviewer_9jpa · 2024-10-22

**Soundness:** 1
**Presentation:** 4
**Contribution:** 2
**Rating:** 3
**Confidence:** 4

**Summary:**

This paper presents a new offline-to-online (O2O) reinforcement learning (RL) method, where they replace the offline RL pre-training
stage, with offline unsupervised RL pre-training.
The offline unsupervised RL pre-training consists of learning an offline policy conditioned on a latent skill vector, resembling a multi-task or goal-conditioned policy.
The paper proposed to name this problem setting as unsupervised-to-online RL (U2ORL).
They evaluate their method in nine tasks from five benchmarks.
They claim that their results suggest that their approach either matches or outperforms O2O RL methods.

**Strengths:**

Overall, I found the paper well-written and easy to follow.
The problem that the authors are working on -- pre-training in RL -- is important and definitely of interest to the community.

For the most part, I found the experiments insightful and thorough.
I particularly like the feature dot product analysis, this is a nice addition to the paper.

**Weaknesses:**

The biggest issue with this paper is that the abstract claims "we empirically demonstrate that U2O RL achieves strong performance that matches or
even outperforms previous offline-to-online RL approaches ...".
However, Figure 12 suggests that O2O outperforms U2O when the offline data only contains expert data.
As such, this claim that U2O either outperforms or matches O2O is false. It is dependent on the type of offline data.
The authors clearly address this in the conclusion but as a reader, I am very disappointed getting to the
end of the paper to find the claim in the abstract is false.
The authors need to update the abstract and correct this claim.

Further to this, I want to know how U2O compares to O2O in Kitchen when using the "complete" data set.
It seems like the authors have avoided including results when using expert data sets because their method
performs worse than O2O RL methods in this setting.
These results are important and should be included.
I suggest the authors include results for the Kitchen task with the complete data set and include a discussion of how their method performs against O2O RL when using different types of offline data sets.

How I see it, this method requires a diverse data set that is collected by an unsupervised RL method.
This paper then proposes a new way to pre-train on this diverse data set.
That is a good contribution, but do not overclaim your contribution.
This could, for example, motivate collecting diverse data sets and the investigation of how to best incorporate expert data into this method.

Finally, this paper is fairly incremental as it simply combines existing methods.
As such, I think it is important that the experiments are thorough so that we gain lots of insights about why we should care about this U2O method.
I think the authors are almost there as the results provide insights that I think are valuable to the community.

**Questions:**

- Do you agree that your method does not outperform O2O methods when using expert data sets?
- Why have you only included one experiment using exert data sets, put it in the appendix, and not explained the results in the main text?

---

> ### Author Response · Authors · 2024-11-20
> **Rebuttal by Authors**
>
> Thank you for the detailed review. We especially appreciate the reviewer’s questions and feedback about some potential overclaims in the paper. Following the suggestion, we have conducted additional experiments with the $\texttt{kitchen-complete}$ dataset, and have revised the paper to avoid any overclaims. Please find our response below. The revisions made are marked with “$\text{\color{red}red}$” in the revised paper.
>
> **Q1. The claim made in the abstract "Do you agree that your method does not outperform O2O methods when using expert data sets?"**
>
> **A1.** Thanks for the feedback. While we have shown that U2O often matches or outperforms O2O across many different tasks, indeed it is not *always* the case, as pointed out by the reviewer and as mentioned in the conclusion section. We agree that the sentence in the abstract is not entirely accurate in its current form, and have toned it down to more accurately reflect our experimental results. Thanks again for pointing out this issue.
>
> Regarding the question, yes, as we have previously mentioned in Section 7, we would agree that U2O would be most effective when the underlying dataset is diverse (so that the agent can learn diverse skills and rich representations), and it might not be as effective when the dataset is monolithically optimal.
>
> **Q2. "I want to know how U2O compares to O2O in Kitchen when using the "complete" data set." / "Why have you only included one experiment using exert data sets, put it in the appendix, and not explained the results in the main text?"**
>
> Thanks for the suggestion! While we have originally mentioned this point as a limitation in the main text (L531-L539), the experiments were limited to two tasks, $\texttt{antmaze-large}$ and $\texttt{antmaze-ultra}$. Hence, following the suggestion, we conducted additional experiments on $\texttt{kitchen-complete}$.
>
> **Normalize return in Kitchen-complete (4 seeds)**
> \begin{array}{l|cccccc}
> \hline
> \text{Environment Steps} & 0.0 \times 10^5 & 1.0 \times 10^5 & 2.0 \times 10^5 & 3.0 \times 10^5 & 4.0 \times 10^5 & 5.0 \times 10^5 \newline
> \hline
> \text{U2O RL (Ours)} & 46.37 \pm 17.60 & 52.75 \pm 17.20 & 52.94 \pm 17.47 & 64.19 \pm 6.49 & 58.56 \pm 11.56 & 59.69 \pm 9.79 \newline
> \text{O2O RL} & 35.63 \pm 21.68 & 48.23 \pm 18.96 & 51.56 \pm 19.73 & 54.69 \pm 11.74 & 48.06 \pm 15.11 & 50.50 \pm 14.87 \newline
> \hline
> \end{array}
>
> We found that, perhaps a bit unexpectedly, U2O slightly outperforms O2O on $\texttt{kitchen-complete}$. Honestly, we don't think this indicates that U2O can necessarily be better than O2O even on some monolithic datasets. Instead, we would interpret this result (and the results in Appendix A.3) as follows: even on monolithic datasets, U2O is not necessarily significantly worse than O2O (though U2O can be a bit slower at the beginning on expert-only datasets for some $\texttt{antmaze}$ tasks). We find this to be rather promising because, unlike traditional O2O, U2O does not use any task information during pre-training, and thus a single pre-trained model can be applied to many different reward functions. Hence, we believe U2O should generally be preferred to O2O if they achieve similar performances.
>
>
> We thank the reviewer again for providing valuable comments and raising important points about U2O RL. We hope that our revisions have removed any potential overclaims in the revised version of the paper. We also believe the additional results with the $\texttt{kitchen-complete}$ dataset have significantly improved the paper. Please let us know if we have addressed your concerns and if so, we would be grateful if the reviewer could consider adjusting the score.

---

> > ### Comment · Reviewer_9jpa · 2024-11-27
> >
> > Thanks for your response and addressing my questions.
> >
> > I find the new results table a bit misleading. You have bolded your method for all tasks whilst the confidence intervals suggest that most of these results are not statistically significant. Only statistically significant results should be bolded, for example, using a paired t-test.

---

> > > ### Author Response · Authors · 2024-12-01
> > > **Response to Reviewer 9jpa**
> > >
> > > Thanks for the response. We modified the Tables so that boldface is applied only when the mean is greater than the mean + standard deviation of the second-best method.

---

### Official Review · Reviewer_rCe7 · 2024-11-04

**Soundness:** 3
**Presentation:** 3
**Contribution:** 2
**Rating:** 3
**Confidence:** 4

**Summary:**

The paper presents an empirical analysis of how unsupervised to online finetuning for RL is better than offline to online finetuning.

**Strengths:**

1. The paper is clearly written and well motivated, having a single reusable model to perform finetuning makes sense and can improve RL pretraining.
2. The paper looks at a particular method HILP and provides a reward scale matching scheme to enable finetuning. This turns out to be quite important in performing efficient finetuning.
3. The paper considers a wide variety of tasks to demonstrate potential improvements over offline to online finetuning.

**Weaknesses:**

1. Insufficient empirical evidence to claim U20>O2O: In figure 3, it seems the results are not significant in 10/14 environments. How can we claim the U20 is a better strategy? Furthermore, insufficient details are provided about baselines of O2O and  off policy RL - eg. do they use the same network sizes and discount factor? It is clear in Table 1 that the baselines and U2O use different network sizes and discount factor as the prior entries are based on discount of 0.99 and use a network size of (256,256) for most tasks. These raises a number of questions on empirical evaluations - maybe the improvements in some domain is because of discount factor?
2. Claim of better features: As I understand, the main claim of the paper is that U2O learn better features. Feature rank collapse is a known problem with offline RL but there have been fixes provided for it in the past. Ex. DR3. It seems comparisons are not made to those modifications at all in this paper.
3. Generality of approach: Unsupervised RL encompasses a broad range of methods. Methods that are based on maximizing mutual information; methods that discover options; methods that capture a bag of skills and update it over time (eg. Voyager). This method relies on reusing policy and value functions to initialize and reward shaping; how will this method work for all the other unsupervised RL approaches. To be calling this a framework might be an overstatement as they consider very related unsupervised approaches based on a single bucket of successor features.

4. Prior works have proposed using unsupervised to online objectives and can be attributed correctly: [1,2,3,4]. I believe the claiming of an entirely new framework is somewhat overclaimed.

[1]:Nasiriany, Soroush, et al. "Planning with goal-conditioned policies." *Advances in neural information processing systems* 32 (2019).

[2]:Eysenbach, Ben, Russ R. Salakhutdinov, and Sergey Levine. "Search on the replay buffer: Bridging planning and reinforcement learning." *Advances in neural information processing systems* 32 (2019).

[3]: Ma, Yecheng Jason, et al. "Vip: Towards universal visual reward and representation via value-implicit pre-training." *arXiv preprint arXiv:2210.00030* (2022).

[4]: Ma, Yecheng Jason, et al. "Liv: Language-image representations and rewards for robotic control." *International Conference on Machine Learning*. PMLR, 2023.

**Questions:**

1. Can the explanation of O2O vs off-policy online RL can be made clear in the paper? Those are important baselines. It seems to be important to put the U2O algorithm in main paper as it wasnt clear that the old dataset was kept around.  It would help to be very clear the differences between U20, O2O and Off-policy online RL
2. Why are results missing in Table 1?
3. In line 249, it might be helpful to have a citation for the reward regression technique with successor features reward.

---

> ### Author Response · Authors · 2024-11-20
> **Rebuttal by Authors (1/2)**
>
> Thank you for the detailed review. We especially appreciate the reviewer’s questions and feedback about the significance of our empirical results and the potential overclaim of the scope. Following the suggestion, we have conducted additional experiments with the DR3 regularizer, and have revised the paper to avoid any potential overclaims. Please find our response below. The revisions made are marked with “$\text{\color{red}red}$” in the revised paper.
>
> **Q1. "Insufficient empirical evidence to claim U20>O2O: In figure 3, it seems the results are not significant in 10/14 environments. How can we claim the U2O is a better strategy?"**
>
> **A1.** Thanks for asking this question! Although U2O outperforms O2O in several tasks, as the reviewer pointed out, U2O indeed does not *always* outperform O2O. However, we would like to highlight one important advantage of U2O: it does **not** use any task information during pre-training, and thus a single pre-trained model can be applied to many different reward functions (unlike U2O), as we have demonstrated in Figure 8. Hence, we believe U2O should generally be preferred to O2O if they achieve similar performances.
>
>
> **Q2. Network sizes and discount factor for U2O and O2O**
>
> **A2.** Thanks for asking for the details about hyperparameters. We would first like to note that, in Figure 3, we ensured a fair comparison between U2O and O2O with the same amount of hyperparameter tuning (e.g., search over {256, 512} for the hidden dimension and {2, 3} for the number of layers, for both methods). In Table 1, we directly took the results from the previous work (Cal-QL), and thus it does not necessarily make a complete apples-to-apples comparison (due to the potential difference in network sizes; though we used the same discount factor of 0.99 for all methods on D4RL tasks). That said, we believe this table still helps contextualize our performance in previously reported numbers. Moreover, for the strongest baseline in Table 1 (Cal-QL), we additionally tuned and ran it ourselves to obtain the results in the most complex environment ($\texttt{antmaze-ultra}$), and found that U2O outperforms Cal-QL by a significant margin.
>
> **Q3. Can DR3 be used to prevent feature collapse in O2O?**
>
> **A3.** Thanks for the question! Following the reviewer's suggestion, we conducted additional experiments to compare U2O and O2O+DR3 on AntMaze-ultra-diverse.
>
> **Normalize return in AntMaze-ultra-diverse (4 seeds)**
> \begin{array}{l|cccccc}
> \hline
> \text{Environment Steps} & 0.0 \times 10^5 & 2.0 \times 10^5 & 4.0 \times 10^5 & 6.0 \times 10^5 & 8.0 \times 10^5 & 10.0 \times 10^5 \newline
> \hline
> \text{U2O RL (Ours)} & \bf{22.00} \pm 9.97 & \bf{36.25} \pm 13.24 & \bf{43.00} \pm 10.09 & \bf{47.00} \pm 24.26 & \bf{46.50} \pm 12.91 & \bf{54.00} \pm 14.24 \newline
> \text{O2O RL + DR3} & 4.67 \pm 5.03 & 18.67 \pm 6.11 & 29.33 \pm 12.22 & 27.33 \pm 10.07 & 30.00 \pm 5.29 & 32.00 \pm 12.49 \newline
> \text{O2O RL} & 13.25 \pm 6.67 & 9.25 \pm 6.67 & 14.25 \pm 11.73 & 23.00 \pm 9.44 & 29.75 \pm 20.27 & 29.00 \pm 13.18 \newline
> \hline
> \end{array}
>
> The result above shows that simply adding the DR3 regularizer is not as effective as U2O RL, and we believe this is likely because the full unsupervised RL procedure can lead to much richer representations than simply adding a regularizer. We have added this result to Appendix A.7 of the revised draft.
>
> **Q4. "To be calling this a framework might be an overstatement as they consider very related unsupervised approaches based on a single bucket of successor features."**
>
> **A4.** Thanks for the feedback. As the reviewer pointed out, and as we have mentioned throughout the paper, U2O RL assumes a *skill-based* offline unsupervised RL method (e.g., successor feature learning, offline goal-conditioned RL, etc.), and is not directly compatible with other types of unsupervised RL methods in the current form. Following the suggestion, and to avoid any overclaiming, we have revised the paper to replace "framework" with **"recipe"**, and have clarified that we assume a skill-based unsupervised offline RL algorithm throughout the paper.

---

> > ### Author Response · Authors · 2024-11-20
> > **Rebuttal by Authors (2/2)**
> >
> > **Q5. "Prior works have proposed using unsupervised-to-online objectives and can be attributed correctly."**
> >
> > **A5.** Indeed, several prior works (including the works mentioned in the review) have considered adapting unsupervised RL for downstream tasks, as we have extensively discussed in the related work section (Section 2 and especially L143-L147). However, to our knowledge, this is the first work that considers the fine-tuning of unsupervised pre-trained skill policies *in the context of offline-to-online RL* (L147-L150). We hope that the clarification in L147-L150 and the replacement of the word "framework" with "recipe" throughout the paper (please refer to our response **A4**) address the reviewer's concern about overclaiming, but please feel free to let us know if there are remaining concerns regarding this point. Finally, we have acknowledged the works the reviewer suggested (LEAP, SoRB, VIP, LIV) in Section 2 of the revised draft.
> >
> > **Q6. "Why are results missing in Table 1?"**
> >
> > **A6.** We took the numbers from previous works (L380), and we left some cells blank if the corresponding numbers were not present in the previous works.
> >
> > **Q7. "In line 249, it might be helpful to have a citation for the reward regression technique with successor features reward."**
> >
> > **A7.** Thanks for the suggestion, we have added references!
> >
> > We thank the reviewer again for providing valuable comments and raising important questions about U2O RL. We believe the additional results with the DR3 regularizer as well as the revisions for avoiding potential overclaims have substantially improved the paper. Please let us know if we have addressed your concerns and if so, we would be grateful if the reviewer could consider adjusting the score.
> >
> > ---
> > **References**
> >
> > [Ma et al., 2023] VIP: Towards Universal Visual Reward and Representation via Value-Implicit Pre-Training. ICLR 2023
> >
> > [Ma et al., 2023] LIV: Language-Image Representations and Rewards for Robotic Control. ICML 2023
> >
> > [Nasiriany et al., 2019] Planning with Goal-Conditioned Policies. NeurIPS 2019
> >
> > [Eysenbach et al., 2019] Search on the replay buffer: Bridging planning and reinforcement learning. NeurIPS 2019

---

> > > ### Comment · Reviewer_rCe7 · 2024-11-25
> > >
> > > Thanks for responding to clarifications.
> > > though we used the same discount factor of 0.99 for all methods on D4RL tasks).
> > > - In Table 3, it is given that Hilbert representation discount factor 0.96 (Walker), 0.98 (others)
> > > It does look the unsupervised part uses different discount factor than the offline rl methods compare. I am not convinced that it is a fair comparison. I also think the novelty of paper lies in empirical design, and without proper controlled experiments, the community can be misled if the comparisons are not accurate.
> > >
> > > Can the authors comment on my Question 1 too? I think it is still unclear in the paper.\
> > > - we believe this is likely because the full unsupervised RL procedure can lead to much richer representations than simply adding a regularizer.
> > >
> > > Again, I think results on one environments is not sufficient to make the case for unsupervised RL learning a better representation. Also there are further training schemes to regularized representation leanring (TD7, Bridging State and History Representations: Understanding Self-Predictive RL, etc). These baselines seem quite important to the paper.
> > >
> > > Further, I still think the novelty of the paper is limited and without controlled experiments I lie on the fence about this work.

---

> > > > ### Author Response · Authors · 2024-12-01
> > > > **Response to Reviewer rCe7**
> > > >
> > > > Thanks again for your comments. We respond to your follow-up comments as follows.
> > > >
> > > > **Q8. Hilbert representation discount factor for DMC environments.**
> > > >
> > > > **A8.** We apologize if our previous response was not entirely clear. We would like to note that **we have ensured a fair comparison:** for *D4RL* tasks, we used an RL discount factor of 0.99 for all methods, and for *DMC* tasks (e.g., Walker), we used a different RL discount factor (0.98), but the same RL discount factor is applied to all methods (U2O, O2O and Online w/ Off Data). We note that the "Hilbert representation discount factor" is a totally *separate* hyperparameter only used to train the $\phi$ representation (which controls the decaying factor in temporal distances), and is independent of the RL discount factor for the underlying RL algorithm.
> > > >
> > > >
> > > > **Q9. Can the explanation of O2O vs off-policy online RL can be made clear in the paper? Those are important baselines. It seems to be important to put the U2O algorithm in main paper as it wasnt clear that the old dataset was kept around. It would help to be very clear the differences between U2O, O2O, and Off-policy online RL**
> > > >
> > > > **A9.** Thanks for the question. While we have reiterated the difference between the three methods in Q1 of Section 5 in the current paper, we will further emphasize the difference between U2O, O2O, and Off-policy online RL in the camera-ready version. In short: they mainly differ only in the offline phase — U2O trains *unsupervised* offline RL, O2O trains *supervised* offline RL, and off-policy online RL does *not* train anything.
> > > >
> > > > **Q10.** Results on one environment is not sufficient to make the case for unsupervised RL learning a better representation. Also, there are further training schemes to regularized representation learning (TD7, Bridging State and History Representations: Understanding Self-Predictive RL, etc). These baselines seem quite important to the paper.
> > > >
> > > > **A10.** Thanks for the comments. Following your suggestion, we provide additional results that compare U2O RL and O2O RL + DR3 with different environments to support that unsupervised RL learns better representations. Moreover, we compare U2O RL with the O2O RL + self-predictive regularizer (SPR) [Ni et al., 2024], which minimizes prediction loss on the next state feature given the current state feature and an action. As shown in the Table below, we find that U2O RL outperforms O2O RL + SPR, which further emphasizes that “supervised” offline RL with a regularizer is not enough. We will add these results in the final draft.
> > > >
> > > > **Normalize return in AntMaze-ultra-diverse (4 seeds)**
> > > > \begin{array}{l|cccccc}
> > > > \hline
> > > > \text{Environment Steps} & 0.0 \times 10^5 & 2.0 \times 10^5 & 4.0 \times 10^5 & 6.0 \times 10^5 & 8.0 \times 10^5 & 10.0 \times 10^5 \newline
> > > > \hline
> > > > \text{U2O RL (Ours)} & \bf{22.00} \pm 9.97 & \bf{36.25} \pm 13.24 & \bf{43.00} \pm 10.09 & \bf{47.00} \pm 24.26 & \bf{46.50} \pm 12.91 & \bf{54.00} \pm 14.24 \newline
> > > > \text{O2O RL + SPR} & 0.00 \pm 0.00 & 0.00 \pm 0.00 & 0.00 \pm 0.00 & 0.00 \pm 0.00 & 0.00 \pm 0.00 & 0.00 \pm 0.00 \newline
> > > > \text{O2O RL + DR3} & 4.67 \pm 5.03 & 18.67 \pm 6.11 & 29.33 \pm 12.22 & 27.33 \pm 10.07 & 30.00 \pm 5.29 & 32.00 \pm 12.49 \newline
> > > > \text{O2O RL} & 13.25 \pm 6.67 & 9.25 \pm 6.67 & 14.25 \pm 11.73 & 23.00 \pm 9.44 & 29.75 \pm 20.27 & 29.00 \pm 13.18 \newline
> > > > \hline
> > > > \end{array}
> > > >
> > > > **Normalize return in Kitchen-partial (4 seeds)**
> > > > \begin{array}{l|cccccc}
> > > > \hline
> > > > \text{Environment Steps} & 0.0 \times 10^5 & 1.0 \times 10^5 & 2.0 \times 10^5 & 3.0 \times 10^5 & 4.0 \times 10^5 & 5.0 \times 10^5 \newline
> > > > \hline
> > > > \text{U2O RL (Ours)} & 47.69 \pm 6.01 & 69.69 \pm 6.84 & 74.38 \pm 0.64 & 74.38 \pm 0.44 & 74.81 \pm 0.37 & 75.00 \pm 0.00 \pm \newline
> > > > \text{O2O RL + DR3} & 29.44 \pm 13.06 & 45.50 \pm 19.78 & 46.50 \pm 26.95 & 62.00 \pm 16.54 & 65.25 \pm 15.92 & 61.88 \pm 23.35 \newline
> > > > \text{O2O RL + SPR} & 37.75 \pm 8.82 & 42.75 \pm 7.29 & 39.38 \pm 4.50 & 42.13 \pm 5.12 & 41.00 \pm 12.31 & 37.00 \pm 3.72 \newline
> > > > \text{O2O RL} & 41.81 \pm 5.56 & 55.50 \pm 8.26 & 69.31 \pm 3.26 & 72.25 \pm 2.73 & 73.63 \pm 2.56 & 74.94 \pm 0.18 \newline
> > > > \hline
> > > > \end{array}
> > > >
> > > > **Normalize return in Adroit-door-binary (4 seeds)**
> > > > \begin{array}{l|cccccc}
> > > > \hline
> > > > \text{Environment Steps} & 0.0 \times 10^5 & 1.0 \times 10^5 & 2.0 \times 10^5 & 3.0 \times 10^5 & 4.0 \times 10^5 & 5.0 \times 10^5 \newline
> > > > \hline
> > > > \text{U2O RL (Ours)} & 22.50 \pm 12.77 & 73.00 \pm 18.55 & \bf{86.75} \pm 10.58 & 90.75 \pm 8.48 & \bf{89.00} \pm 15.68 & 93.00 \pm 7.71 \newline
> > > > \text{O2O RL + DR3} & 28.00 \pm 18.76 & 51.50 \pm 34.81 & 48.50 \pm 18.21 & 82.50 \pm 14.27 & 75.50 \pm 13.30 & 88.50 \pm 11.12 \newline
> > > > \text{O2O RL + SPR} & 34.50 \pm 10.25 & 44.00 \pm 14.70 & 45.00 \pm 20.30 & 64.50 \pm 21.13 & 45.00 \pm 19.49 & 62.50 \pm 15.26 \newline
> > > > \text{O2O RL} & 30.00 \pm 6.69 & 54.33 \pm 15.36 & 67.67 \pm 16.46 & 76.67 \pm 14.40 & 73.00 \pm 15.74 & 79.00 \pm 12.25 \newline
> > > > \hline
> > > > \end{array}

---

### Official Review · Reviewer_oCcT · 2024-11-04

**Soundness:** 2
**Presentation:** 3
**Contribution:** 2
**Rating:** 5
**Confidence:** 3

**Summary:**

This paper introduces Unsupervised-to-Online Reinforcement Learning (U2O RL) as an alternative to the conventional Offline-to-Online RL (O2O RL) framework. U2O RL replaces domain-specific, supervised offline RL with unsupervised offline RL, allowing a single pre-trained model to be adapted for multiple tasks. Experiments conducted across nine environments demonstrate that U2O RL can match or even surpass previous methods.

I do not currently agree with the direct acceptance of this paper. However, I recognize the importance of research on RL in pretrain-finetune frameworks. If the authors can satisfactorily address my concerns, I would be open to increasing my rating.

**Strengths:**

- The paper is well-written, and the proposed U2O RL framework is explained clearly.
- Research into specific paradigms within the pretrain-finetune framework for RL is valuable, and this paper contributes to that discussion.

**Weaknesses:**

- While this paper proposes the U2O RL framework, it does not introduce any novel methods. Both the unsupervised offline RL pre-training and the online fine-tuning stages rely on existing algorithms. Additionally, the reward scaling adjustment in the bridging stage has already been employed in prior reward design approaches. Proposing a new “framework” is reasonable, but I believe the paper needs to provide more substantial evidence on why this U2O RL framework is more effective than the traditional O2O approach. Simply demonstrating feature co-adaptation to support the efficacy of representation learning in the offline phase may be insufficient. Furthermore, the paper should conduct a broader set of experiments: for instance, by testing various offline and online algorithms and using a wider range of environments to validate the framework’s effectiveness. Without these enhancements, the work seems somewhat incremental.
- For general reward maximization tasks, such as Walker, Cheetah, and Jaco, the U2O RL framework does not demonstrate a notable advantage; in some cases (e.g., Cheetah), performance is even lower. This could stem from difficulties in selecting an optimal skill latent vector  $z^*$  for these tasks, highlighting a potential limitation of the U2O framework.

**Questions:**

- Why does the feature dot product in the O2O framework for Walker Run and Cheetah Run (Fig. 4) diverge, indicating much poorer representation learning compared to U2O, yet the performance of O2O in these environments is similar or even superior to U2O? Intuitively, poorer representations should lead to poorer performance.

---

> ### Author Response · Authors · 2024-11-20
> **Rebuttal by Authors (1/2)**
>
> Thank you for the detailed review. We especially appreciate the reviewer’s question about the contribution of our work. Following the suggestion, we have conducted additional experiments with manipulation tasks. Please find our response below. The revisions made are marked with “$\text{\color{red}red}$” in the revised paper.
>
> **Q1. "While this paper proposes the U2O RL framework, it does not introduce any novel methods. Both the unsupervised offline RL pre-training and the online fine-tuning stages rely on existing algorithms."**
>
> **A1.** Thanks for raising the question about our contributions. As the reviewer pointed out, individual components of U2O RL are not necessarily completely novel. However, to our knowledge, we are the first to show that (skill-based) unsupervised offline RL can often be even better than supervised offline RL for online fine-tuning *even for the same task* (L087). We believe this observation is novel (at least it was quite surprising to us — how can *not* using task information during pre-training possibly be better than using it?). In the paper, we experimentally identified the potential cause behind this phenomenon (i.e., representation quality issue) via our analyses, and developed a practical method (U2O RL) to substantiate this insight. We believe our insight and practical recipe constitute a solid contribution to the community.
>
> However, we agree with the reviewer that calling it an entirely novel framework might potentially be a bit misleading. We have thus revised the manuscript to call it a **"recipe"** instead of a "framework", and toned down several claims accordingly.
>
> **Q2. “The paper should conduct a broader set of experiments.”**
>
> **A2.** Thanks for the feedback. To further strengthen the results, we additionally conducted experiments with new manipulation environments ($\texttt{cube-single}$ and $\texttt{cube-double}$) from OGBench [Park et al., 2024], bringing the total to $\mathbf{11}$ environments. The goal of these cube tasks is to perform pick-and-place manipulation of multiple cubes from an unlabeled dataset that consists of diverse, random pick-and-place behaviors.
>
> **Success rate in OGBench-cube-single (4 seeds)**
> \begin{array}{l|cccccc}
> \hline
> \text{Environment Steps} & 0.0 \times 10^5 & 2.0 \times 10^5 & 4.0 \times 10^5 & 6.0 \times 10^5 & 8.0 \times 10^5 & 10.0 \times 10^5 \newline
> \hline
> \text{U2O RL (Ours)} & \bf{72.50} \pm 12.79 & \bf{90.00} \pm 9.38 & \bf{97.00} \pm 1.00 & \bf{99.00} \pm{1.15} & \bf{100.00} \pm 0.00 & \bf{99.50} \pm 1.00 \newline
> \text{O2O RL} & 36.50 \pm 9.43 & 79.00 \pm 7.75 & 71.00 \pm 4.76 & 78.00 \pm 9.38 & 91.50 \pm 5.74 & 94.50 \pm 4.43 \newline
> \hline
> \end{array}
>
>
> **Success rate in OGBench-cube-double (4 seeds)**
> \begin{array}{l|cccccc}
> \hline
> \text{Environment Steps} & 0.0 \times 10^5 & 2.0 \times 10^5 & 4.0 \times 10^5 & 6.0 \times 10^5 & 8.0 \times 10^5 & 10.0 \times 10^5 \newline
> \hline
> \text{U2O RL (Ours)} & \bf{2.50} \pm 5.00 & \bf{21.00} \pm 14.09 & \bf{19.00} \pm 13.61 & \bf{33.50} \pm 19.82 & \bf{38.00} \pm 21.97 & \bf{44.00} \pm 27.52 \newline
> \text{O2O RL} & 0.50 \pm 1.00 & 0.50 \pm 1.00 & 0.00 \pm 0.00 & 0.00 \pm 0.00 & 0.50 \pm 1.00 & 0.00 \pm 0.00 \newline
> \hline
> \end{array}
>
> The tables above show that U2O outperforms O2O in these complex tasks as well. We have added these new results to the revised draft.
>
> **Q3. "the paper needs to provide more substantial evidence on why this U2O RL framework is more effective than the traditional O2O approach"**
>
> **A3.** Through our experiments, we showed that U2O often matches or outperforms O2O across a variety of domains and that it often learns representations of high quality. Unfortunately, U2O does not *always* outperform O2O; indeed, it sometimes just achieves a matching performance to O2O, as the reviewer mentioned. However, we would like to highlight one important advantage of U2O: it does **not** use any task information during pre-training, and thus a single pre-trained model can be applied to many different reward functions (unlike U2O), as we have demonstrated in Figure 8. Hence, we believe U2O should generally be preferred to O2O if they achieve similar performances.
>
> **Q4. "Why does the feature dot product in the O2O framework for Walker Run and Cheetah Run (Fig. 4) diverge, indicating much poorer representation learning compared to U2O, yet the performance of O2O in these environments is similar or even superior to U2O?"**
>
> **A4.** Thanks for this valid question! As briefly noted in the paper (L460) (and as shown in previous work [Fu et al., 2022]), we believe the quality of representations *alone* does not fully explain the performance, although there exists a correlation to some degree [Kumar et al., 2022]. In Figure 4, we mainly wanted to highlight that the representation quality of U2O is generally better than that of O2O, which we believe can partially explain the performance of U2O in some environments.

---

> > ### Author Response · Authors · 2024-11-20
> > **Rebuttal by Authors (2/2)**
> >
> > We thank the reviewer again for providing valuable comments and feedback. Please let us know if we have addressed your concerns and if so, we would be grateful if the reviewer could consider adjusting the score.
> >
> > ---
> >
> > **References**
> >
> > [Park et al., 2024] OGBench: Benchmarking Offline Goal-Conditioned RL. arXiv 2024
> >
> > [Fu et al., 2022] A Closer Look at Offline RL Agents. NeurIPS 2022
> >
> > [Kumar et al., 2022] DR3: Value-Based Deep Reinforcement Learning Requires Explicit Regularization. ICLR 2022

---

> ### Comment · Reviewer_oCcT · 2024-11-29
>
> Thanks for the detailed responses.
>
> Although I believe that research on pretraining-finetuning in RL is quite important, I think this paper does not adequately explain when the proposed U2O is more applicable, because clearly U2O does not show advantages in all environments. Additionally, the data available for pretraining might be limited to that environment or related environments, which is quite different from LLM pretraining where data from a wide range of unrelated tasks can be utilized. I believe that U2O still faces significant challenges in terms of broader applicability. Furthermore, I suggest that the authors provide some recommendations on the types of environments or tasks where U2O can perform better, and where it might not.
>
> In summary, I regret that I cannot increase my score.

---

> > ### Author Response · Authors · 2024-12-01
> > **Response to Reviewer oCcT**
> >
> > Thanks again for your comments. We respond to your follow-up comments as follows.
> >
> > **Q5.** When does U2O RL perform better than O2O RL?
> >
> > **A5.** Thanks for the question. We would like to note that we already have a separate section for this exact question (Q6 of Section 5 in the updated PDF, which was previously in Section 6 in the submission version). In short, we believe U2O RL should perform better than O2O RL when the offline dataset consists of *diverse* trajectories, and we don't expect U2O to be necessarily better than O2O on expert-only datasets. Since it is often prohibitive to collect expert-only trajectories [Lynch et al., 2020], we believe U2O can enable unsupervised pre-training on much broader scenarios even with suboptimal/task-irrelevant datasets, as we have experimentally demonstrated in the paper.
> >
> > [Lynch et al., 2020] Lynch, C., Khansari, M., Xiao, T., Kumar, V., Tompson, J., Levine, S., & Sermanet, P. Learning latent plans from play. CoRL 2020

---

### Meta-Review · Area_Chair_jg4H · 2024-12-19

**Metareview:**

This paper proposes a recipe for doing unsupervised-to-online RL. The idea is that if you have a big dataset of observation-action pairs from different tasks, you can learn a task-conditioned policy $\pi(a \mid s, z)$ where your $z$ is your task vector. No rewards are used during this part. Then given a task, you can identify the right task vector $z^\star$ using reward values, and do online RL to further improve the performance of the policy $\pi(a \mid s, z^\star)$.

The main strengths of the paper are:

- RL is sample expensive, so pre-training before doing online RL is an important task.
- Approach is simple and easy and also composable (e.g., one can use any online RL approach).


Main weakness:

- The main issue, as raised by reviewers, concerns comparison with O2O. As pointed out by reviewers, U2O does not uniformly outperform offline-to-online (O2O) works across the board and either matches O2O in some domains or even gets outperformed by O2O when the offline data is of expert type. The authors agree with this. The main counterpoint is that expert data is hard to collect and U2O uses a single pre-trained model which is still an advantage. I agree with the latter point, but one would still wish for U2O to match O2O when the latter has expert data.

- Reviewers have mentioned that the paper overclaims its contribution of a general framework when the approach is more specific and also overclaims gains over past O2O work in the abstract. In response, the authors have edited the writing to reduce their claims.

- Reviewers have also mentioned that the approach isn't novel. However, this in itself doesn't concern me. In fact, it is great to reuse existing approaches. But what would have then been good is to see a more real-world application rather than simulator-based evaluations.

The authors have addressed these claims by removing overclaims of a framework, toning down the abstract, and adding additional experiments. I worry that this is a non-trivial amount of change that should be another submission. Further, while the approach is simple, neither it is strictly better than the past approach, nor is the core idea of using pre-training for RL novel, nor it is demonstrated in a real-world application. For this reason, I am currently leaning towards rejection but I won't mind if it gets accepted.

**Additional Comments On Reviewer Discussion:**

Reviewers raised the following main concerns

1. U2O does not outperform past work O2O uniformly and in fact underperforms when there is offline expert data
2. Overclaim in claiming a framework
3. Lack of novelty

Of this, only (1) and (2) worries me. I am not too worried about (3), in fact, simpler approaches should be preferred over forced novelty. However, what is missing here is a real-world application that shows U2O is helpful. Unfortunately, this is true for most research in offline RL.

---

### Decision · Program_Chairs · 2025-01-22

Reject